# RIEMANN-LEBESGUE FOREST FOR REGRESSION

## ABSTRACT

We propose a novel ensemble method called Riemann-Lebesgue Forest (RLF) for regression. The core idea in RLF is to mimic the way how a measurable function can be approximated by partitioning its range into a few intervals. With this idea in mind, we develop a new tree learner named Riemann-Lebesgue Tree (RLT) which has a chance to perform Lebesgue type cutting,i.e splitting the node from response $Y$ at certain non-terminal nodes. In other words, we introduce the "splitting type randomness" in our ensemble method. We show that the optimal Lebesgue type cutting results in larger variance reduction in response $Y$ than ordinary CART (Breiman et al., 1984) cutting (an analogue of Riemann partition). Such property is beneficial to the ensemble part of RLF. We also generalize the asymptotic normality of RLF under different parameter settings. Two one-dimensional examples are provided to illustrate the flexibility of RLF. The competitive performance of RLF against original random forest (Breiman, 2001) is demonstrated by experiments in simulation data and real world datasets.

## 1 INTRODUCTION

Random Forest (Breiman, 2001) has been a successful ensemble method in regression and classification tasks for decades. Combination of weak decision tree learners reduces the variance of a random forest (RF) and results in a robust improvement in performance. "Feature bagging" further prevents RF from being biased toward strong features which might cause fitted subtrees are correlated. However, the benefit of "feature bagging" may be limited when only small proportion of features are informative. In that case, RF is likely to learn noisy relationship between predictors and response which in the end makes RF underfit the true functional relationship hidden in the data.

Many methods have been proposed to tackle this issue. Heaton (2016) proposed to rule out irrelevant features or perform feature engineering at the beginning of fitting a RF . Another type of ideas is to adjust the way RF selecting informative features. Amaratunga et al. (2008), Ghosh & Cabrera (2021) employed weighted random sampling in choosing the eligible features at each node. Besides the feature weighting method, Xu et al. (2012) used different types of trees such as C4.5, CART, and CHAID to build a hybrid forest. The resulted forest performs well in many high-dimension datasets. Zhou & Feng (2017) employed a sequential multi-grained scanning to discover the feature relationships.

Most of those methods only deal with classification tasks. In this paper, we fill in the gap by proposing a novel forest named Riemann-Lebesgue Forest (RLF) which has superior performance than ordinary Random Forest in regression tasks. The main idea of RLF is to exploit information hidden in the response rather than predictors only. In most types of tree algorithms, people approximate the regression function in the "Riemann" sense. That means the fitted function $\hat{f}(\mathbf{x})$ can be written as follows:

$$\hat{f}(\mathbf{x}) = \sum_{i=1}^{P} \bar{y}_{R_i} \mathbf{1}_{\{\mathbf{x} \in R_i\}} \tag{1}$$

where each $R_i$ is a hypercube in feature space, $\bar{y}_{R_i}$ represents the mean value of responses lying in the region $R_i$ and $P$ is the total number of hypercubes partitioned. Fig.1(a) gives an example of a smooth function fitted by step functions in one dimension.

As we can see, partitioning x-axis may underfit the true function unless we keep partitioning, which means we take the limit of step functions. But that is nearly undoable in practice. Many decision tree

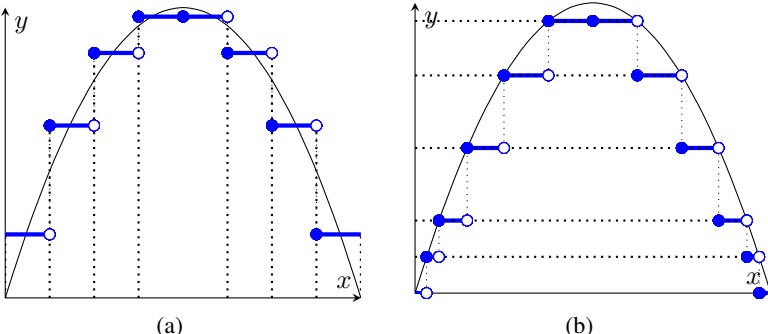

Figure 1: Two types of function approximation.(a):"Riemann" type approximation. (b): "Lebesgue" type approximation

algorithms follow this idea by choosing optimal splitting value of optimal feature at each non-terminal node. On the other hand, we know that any measureable function $f : \mathbb{R}^p \to \mathbb{R}$ can be approximated by a sequence of simple functions $\phi_n(\mathbf{x})$, $n = 1, 2, ....$ . Equation equation 2 gives an example of simple functions:

$$\phi_n(\mathbf{x}) = \sum_{j=1}^{n2^n} \frac{j-1}{2^n} \chi_{A_{j,n}} + n\chi_{B_n} \tag{2}$$

where $A_{j,n} = f^{-1}([\frac{j-1}{2^n}, \frac{j}{2^n}))$ for $j = 1, 2, ..., n2^n$ and $B_n = f^{-1}([n, \infty])$.

It is now a standard analysis exercise to show that $\phi_n$ converges to $f$. Note that if $f$ is finite (which is typically a case in practice), $\chi_{B_n}$ will vanish for large $n$. In other words, we can actually obtain the partitioned feature space indirectly by partitioning the response space (real line) in regression tasks. For comparison, we borrow the term in analysis and name this kind of partition procedure as "Lebesgue" type. Fig. 1(b) gives an example of approximating a function in "Lebesgue" sense. One characteristic for partitioning feature space from response is, the shape of resulted region is not limited to hypercube, which actually enriches the structure of trees in a forest. To overcome the limited structures learned by ordinary RF in sparse models, we incorporate the idea of "Lebesgue" partition in constructing each decision tree. To implement the "Lebesgue" type splitting rule as shown in part (b) in Fig.1(b) , we can apply the CART-split criterion on response $Y$ which is simple but efficient.

The remaining sections of this paper are organized as following schema. In section 2.1, we illustrate the idea of Riemann-Lebesgue Forest (RLF) in detail. In section 3, we present the potential of Lebesgue cutting in reducing the variance of response $Y$. Theoretical results such as asymptotic normality of RLF and time complexity analysis are included. We compare the performance of RLF and RF in sparse model and benchmark real datasets in section 4. Simulation results of tuned RLF in models with small signal-to-noise ratio and mixture distribution will be demonstrated as well. Section 5 discusses few characteristics and limitations of RLF and proposes future directions. The R codes for the implementation of RLF, selected real datasets and the simulation results are available in supplementary materials. All simulations and experiments are performed on a ThinkPad with Intel(R) Core(TM) i5-8250U CPU @ 1.60GHz and 32 GB RAM.

## 2 METHODOLOGY

In section 2.1, we first introduce essential preliminary used in the rest of the paper. We will illustrate how to incorporate "Lebesgue" partition in constructing a CART type tree in section 2.2. Then we apply this new type of tree to the algorithm of Riemann-Lebesgue Forest in section 2.3.

## 2.1 PRELIMINARY

In this paper, we only consider the regression framework where the data consist of $n$ i.i.d pairs of random variables $Z_i = (\mathbf{X}_i, Y_i) \in \mathcal{X} \times \mathbb{R}$, $i = 1, 2, ..., n$. Let $F_Z$ be the common distribution of $Z_i$. WLOG, we can assume the feature space $\mathcal{X} = [0, 1]^d$, where $d$ is the dimension of feature space.

The procedure of generating a Random Forest regression function from Breiman's original algorithm can be summarized as follows:

1. Bootstrapping $N$ times from original dataset.
2. For each bootstrapped dataset, build a CART decision tree based on those subsamples.
3. Take the average of $N$ built trees as the ensemble regressor.

At bootstrapping time $i$, denote $(X_{i1}, Y_{i1}), ..., (X_{ik}, Y_{ik})$ be a corresponding subsample of the training set. If we write the corresponding grown CART tree as $T^{(\omega_i)}((X_{i1}, Y_{i1}), ..., (X_{ik}, Y_{ik}))$, then the final random forest estimator $R_n$ evaluated at point $\mathbf{x}^*$ can be written as:

$$R_n(\mathbf{x}^*) = \frac{1}{N} \sum_{i=1}^{N} T_{\mathbf{x}^*}^{(\omega_i)}((X_{i1}, Y_{i1}), ..., (X_{ik}, Y_k)) \tag{3}$$

where random variables $\omega_i$ represents the randomness from the construction of $i$-th CART tree. For growing a CART tree, at each node of a tree, the CART-split criterion is applied. $m_{try}$ features will be randomly picked as potential splitting directions. Typically $m_{try} = \lfloor d/3 \rfloor$. On the other hand, the structure of a single CART tree in original forests relies on both predictors and responses in the corresponding subsample, i.e the randomness also comes from resampling (bootstrapping procedure). By assumption in Breiman (2001), $(\omega_i)_{i=1}^{n}$ are i.i.d and independent of each bootstrapped dataset.

A detailed description of the CART-split criterion is as follows. Let $A$ be a generic non-terminal node in a decision tree and $N(A)$ be the number of observations lying in node $A$. A candidate cut is a pair $(j, z)$ where j represents a dimension in $\{1, ..., d\}$ and $z$ is the splitting point along the $j$-th coordinate of feature space. Since CART-split criterion in RF focuses on splitting predictors, we can view such cutting method as the "Riemann" type splitting rule. Let the set $\mathcal{C}_R = \{(j, z^{(j)}) : j \in \{p_1, ..., p_{m_{try}}\}, z^{(j)} \in \mathbf{X}^{(j)}\}$ consist of all possible cuts in node $A$ and denote $\mathbf{X}_i = (\mathbf{X}_i^{(1)}, ..., \mathbf{X}_i^{(d)})$, where the set $\{p_1, ..., p_{m_{try}}\}$ represents randomly picked feature index in CART, then for any $(j, z) \in \mathcal{C}_R$, the CART-split criterion (Breiman et al., 1984) chooses the optimal $(j^*, z^*)$ such that

$$(j^*, z^*) \in \underset{\substack{j \in m_{try} \\ (j,z) \in \mathcal{C}_R}}{\arg \max} L(j, z)$$

where

$$L(j, z) = \frac{1}{N(A)} \sum_{i=1}^{n} (Y_i - \bar{Y}_A)^2 \mathbf{1}_{\mathbf{X}_i \in A} - \frac{1}{N(A)} \sum_{i=1}^{n} (Y_i - \bar{Y}_{A_L} \mathbf{1}_{\mathbf{X}_i^{(j)} < z} - \bar{Y}_{A_R} \mathbf{1}_{\mathbf{X}_i^{(j)} \geq z})^2 \mathbf{1}_{\mathbf{X}_i \in A} \tag{4}$$

, $A_L = \{\mathbf{x}_i \in A : \mathbf{x}_i^j < z, i = 1, 2, ..., n\}$, $A_R = \{\mathbf{x}_i \in A : \mathbf{x}_i^j \geq z, i = 1, 2, ..., n\}$ and $\bar{Y}_A, \bar{Y}_{A_L}, \bar{Y}_{A_R}$ are the averages of responses $Y_i$ with the corresponding features are in sets $A, A_L$ and $A_R$, respectively.

## 2.2 RIEMANN-LEBESGUE TREE

To implement the "Lebesgue" type splitting rule as shown in part (b) in Fig.1(b) , we can apply the CART-split criterion on response $Y$. Note that we only need to choose the optimal splitting point for $Y$ in this case. Denote $\mathcal{C}_L = \{(j, z) : j \in \{0\}, z \in Y\}$ be the set consist of all possible splitting points for response, then we choose the optimal splitting point $z_L^*$ at node $A$ such that

$$z_L^* \in \underset{z \in \mathcal{C}_L}{\arg \max} \tilde{L}(z)$$

where

$$\tilde{L}(z) = \frac{1}{N(A)} \sum_{i=1}^{n} (Y_i - \bar{Y}_A)^2 \mathbf{1}_{Y_i \in A} - \frac{1}{N(A)} \sum_{i=1}^{n} (Y_i - \bar{Y}_{\tilde{A}_D} \mathbf{1}_{Y_i < z} - \bar{Y}_{\tilde{A}_U} \mathbf{1}_{Y_i \geq z})^2 \mathbf{1}_{Y_i \in A} \tag{5}$$

, $\tilde{A}_D = \{\mathbf{x}_i \in A : Y_i < z, i = 1, 2, ..., n\}$, $\tilde{A}_U = \{\mathbf{x}_i \in A : Y_i \geq z, i = 1, 2, ..., n\}$ and $\bar{Y}_A, \bar{Y}_{A_D}, \bar{Y}_{A_U}$ are the averages of responses $Y_i$ with the corresponding features are in sets $A, \tilde{A}_D$ and $\tilde{A}_U$, respectively.

As we can see from equation 4 and equation 5, the "Lebesgue" type splitting rule will go through all potential cutting points for $y$ while the "Riemann" type splitting can only check part of them. We can conclude that $L(j^*, z^*) \leq \tilde{L}(z_L^*)$.

One issue for the "Lebesgue" type splitting rule is overfitting. Suppose the response $Y$ at $A'$ only takes two distinct values, say $Y = 0$ or $Y = 1$, on a node $A'$. The CART-split criterion for $Y$ can give a perfect split $z_L^* = 0.5$ under some appropriate sets of $\mathcal{C}_L$. In other words, we will have $\tilde{L}(z^*) = \frac{1}{N(A')} \sum_{i=1}^{n} (Y_i - \bar{Y}_{A'})^2 \mathbf{1}_{Y_i \in A'}$. This phenomenon restricts the potential application of "Lebesgue" partitioning in classification task.

To overcome the potential overfitting from the "Lebesgue" type splitting, we apply "Riemann" and "Lebesgue" splitting in a hybrid way. Since we will eventually construct an ensemble learner from Riemann-Lebesgue trees, it's acceptable to introduce a Bernoulli random variable $B$ to determining splitting types at each non-terminal node $A$. More specifically,

$$B \sim Bernoulli(\tilde{p}), \quad \tilde{p} = \frac{\tilde{L}(z_L^*)}{L(j^*, z^*) + \tilde{L}(z_L^*)}$$

If $B = 1$, the "Riemann" type splitting will be employed, and vice versa. The reason why we define $\tilde{p}$ as above is to control the number of nodes taking "Lebesgue" type splitting. We already seen that $L(j^*, z^*) \leq \tilde{L}(z_L^*)$, so it's expected that there won't be too many "Lebesgue" type nodes and $\tilde{p}$ will play a role of regularization.

Another issue for the Riemann-Lebesgue Tree is the prediction. When there comes a new point $(\mathbf{x}, y)$, we are not allowed to use the information of its response $y$. That enforces us to estimate the value of response locally when we need to know which partition the new point belongs to.

Linear regression is one of the candidates for the local model. However, it is unstable when the sample size is relatively small. Another choice is the K-Nearest Neighborhood algorithm (KNN). However, the performance of KNN relies on the distance function we used which can be unstable in high-dimensional cases.

In this paper, we choose random forest as the local model to obtain an estimate of the response value of a new incoming point since random forest is parameter free and robust for small sample size. We believe there exists more efficient local models, which is our future work. Algorithm 1 summarizes the procedure of fitting a Riemann-Lebesgue Tree. To our best knowledge, this is the first type of base learner exploring information directly from response.

### 2.3 RIEMANN-LEBESGUE FOREST

Once we establish the way to build a Riemann-Lebesgue Tree, the ensemble version follows immediately. We follow the spirit of original Forest (Breiman, 2001) . That is, growing $M$ trees from different subsamples. Each tree is grown as illustrated in Algorithm 1. We employ sampling without replacement (subagging) in ensembling part.For completeness we provide the procedure of predicting $\mathbf{x}$ value from a Riemann-Lebesgue Forest (RLF) in Algorithm 2 of section A.3.

## 3 THEORETICAL ANALYSIS OF RLF

For the sake of theoretical analysis, we give a theoretical version of Riemann-Lebesgue Forest. Suppose $(Z_1, ..., Z_n)$ are i.i.d from a common underlying distribution $F_Z$, where $Z_i = (X_i, Y_i)$. Let $h^{(\omega_i)}$ be the random kernel corresponding to the randomness $\omega_i$ induced by $i$-th subsample [1], where $(\omega_i)_{i=1}^{n}$ are i.i.d with $F_\omega$ and independent of each subsample. Denote $N$ be subagging times and $k$ be subagging size. Since we uniformly samples $k$ distinct data points with replacement, the incomplete $U$-statistic with random kernel (See detailed explanations in section A.4) at the query point $\mathbf{x}$ is

---

[1]More specifically, the randomness of a Riemann-Lebesgue tree given a fixed subsampling comes from the feature bagging and random choice of Riemann type cutting and Lebesgue type cutting.

---

**Algorithm 1** Riemann-Lebesgue Tree (Fitting)

---

**Require:** Training data $\mathcal{D}_n = \{(\mathbf{X}_i, Y_i), i = 1, ..., n\}$. Minimum node size $M_{node}$. $m_{try} \in \{1, 2, ..., d\}$. Let $N(A)$ denote the number of sample points at node $A$ and $A$ is the root of the tree at the beginning. $M_{local}$ is the number of trees used in local random forests.

1: **if** $N(A) > M_{node}$ **then**
2:      Select $M_{try} \subset \{1, 2, ..., d\}$ of cardinality $m_{try}$ without replacement uniformly.
3:      Perform CART-split criterion among the selected features, and obtain $j^*, z^*, L(j^*, z^*)$.
4:      Perform CART-split criterion with respect to $Y_i \in A$, and obtain $z_L^*, \tilde{L}(z_L^*)$.
5:      Calculate $\tilde{p} = \frac{\tilde{L}(z_L^*)}{L(j^*, z^*) + \tilde{L}(z_L^*)}$ and sample $B \sim \text{Bernoulli}(\tilde{p})$.
6:      **if** $B = 1$ **then**
7:          Cut the node $A$ according to $j^*, z^*$. Denote $A_L$ and $A_R$ as the two resulting nodes.
8:          Repeat steps $1 - 14$ for nodes $A_L$ and $A_R$.
9:      **else**
10:         Cut the node $A$ according to $z_L^*$. Denote $A_D$ and $A_U$ as the two resulting nodes.
11:         Fit a random forest model with $M_{local}$ trees w.r.t points in current node $A$. Call it $RF_{local}$.
12:         Repeat steps $1 - 14$ for nodes $A_D$ and $A_U$.
13: **else**
14:      Set the current node $A$ as a terminal node.
15: **Return:** A collection of nodes and fitted CART-splitting rules

---

$$U_{n,k,N;\omega}(\mathbf{x}) = \frac{1}{N} \sum_{i=1}^{N} h^{(\omega(i))}(\mathbf{x}; Z_{i_1}, ..., Z_{i_k})$$

$$= \frac{1}{\binom{n}{k}} \sum_{(n,k)} \frac{W_i}{p} h^{(\omega(i))}(\mathbf{x}; Z_{i_1}, ..., Z_{i_k}) \tag{6}$$

where $p = N/\binom{n}{k}$ and vector

$$W = (W_1, ..., W_{\binom{n}{k}}) \sim Multinomial(N, \frac{1}{\binom{n}{k}}, ..., \frac{1}{\binom{n}{k}})$$

Note that in asymptotic theory, both $N$ and $k$ can rely on sample size $n$. We first show Lebesgue cuttings induce smaller $L_2$ training error than Riemann cutting in section 3.1. In section 3.2, we further give the convergence rate of the asymptotic normality of RLF. To see the time efficiency of RLF in different sizes of data, a complexity analysis is given in section 3.3. Since the Lebesgue part of each Riemann-Lebesgue Tree is essentially splitting the response $Y$ with CART-criterion, many consistency results for traditional RF can be applied to RLF directly. For completeness, in section A.5 of Appendix, we provide a version of consistency adapted to RLF according to the results of Scornet et al. (2014).

## 3.1 VARIANCE REDUCTION OF RESPONSE $Y$ BY LEBESGUE CUTTINGS

In section 2.2, we conclude that $L(j^*, z^*) \leq \tilde{L}(z_L^*)$ for each partition in constructing a Riemann-Lebesgue Tree. Theorem 3.1 formalizes the above conclusion.

**Theorem 3.1.** *Let the regression function be $Y = f(\mathbf{X}) + \varepsilon$ ,where $\mathbf{X} \in \mathbb{R}^d$, $Y \in \mathbb{R}$ and $f$ is a bound measurable function and $\varepsilon$ is the noise term. Under the procedure defined in equation 4 and equation 5, let $A_1^* = \{Y > a^*\}, A_2^* = \{Y \leq a^*\}$ be the optimal Lebesgue cutting and $B_1^* = \{X^{(j^*)} > b^*\}, B_2^* = \{X^{(j^*)} \leq b^*\}$ be the optimal Riemann(CART) cutting ,then we have:*

$$Var(Y) - Var(Y|A_1^*)P(A_1^*) - Var(Y|A_2^*)P(A_2^*) \geq Var(Y) - Var(Y|B_1^*)P(B_1^*) - Var(Y|B_2^*)P(B_2^*)$$

In other words, the variance reduction of response $Y$ induced by the optimal Lebesgue cutting will be greater or equal to that of optimal Riemann cutting. It follows immediately that a RLT will have smaller $L_2$ training error than a ordinary CART tree given other tree parameters are the same. According to the bias-variance decomposition, RLF is expected to have smaller mean squared error than RF after the ensembling step. Real-world data experiments in section 4 verified this conjecture. The proof of Theorem 3.1 is based on the variance decomposition formula (law of total variance).

## 3.2 CONVERGENCE RATE OF THE ASYMPTOTIC NORMALITY

Peng et al. (2019) provided sharp Berry-Esseen bounds for of RF under the Bernoulli sampling (Chen & Kato, 2019). The main idea follows from the Stein's method (Chen et al., 2010) and Hoeffding decomposition (Vaart, 2000). Getting inspired by the results in (Peng et al., 2019) and (Mentch & Hooker, 2016), we derive improved Berry-Esseen bounds of RLF for small-$N$ settings (i.e, relatively small number of trees in RLF) where $\lim \frac{n}{N} = \alpha$ and $\alpha > 0$ or $\infty$ in Theorem 3.2.

**Theorem 3.2.** *Suppose $Z_1, ..., Z_n \overset{i.i.d}{\sim} \mathcal{P}_{\mathbf{Z}}$ and $U_{n,k,N;\omega}$ is defined as in equation 6 with random kernel $h^{(\omega)}(Z_1, ..., Z_k)$. Denote $\theta = \mathbb{E}[h(Z_1, ..., Z_k; \omega)]$. Let $\zeta_k = var(h(Z_1, ..., Z_k; \omega))$, $\zeta_{1,\omega} = \mathbb{E}[g^2(Z_1)]$ where $g(z) = \mathbb{E}[h(z, Z_2, ..., Z_k; \omega)] - \theta$. And $p = N/\binom{n}{k}$. Denote $\alpha_n = \frac{n}{N}$. If $\zeta_k < \infty, \zeta_{1,\omega} > 0, 0 < \eta_0 < 1/2$ and $\mathbb{E}[|h - \theta|^{2m}]/\mathbb{E}^2[|h - \theta|^m]$ is uniformly bounded for $m = 2, 3$, we have the following Berry-Esseen bound for $U_{n,k,N;\omega}$:*

$$
sup_{z \in R} \left| P\left( \frac{\sqrt{N}(U_{n,k,N;\omega} - \theta)}{\sqrt{k^2 \zeta_{1,\omega}/\alpha_n + \zeta_k}} \le z \right) - \Phi(z) \right| \le \tilde{C} \Bigg\{ \frac{\mathbb{E}|g|^3}{n^{1/2} \zeta_{1,\omega}^{3/2}} + \left[ \frac{k}{n} \left( \frac{\zeta_k}{k\zeta_{1,\omega}} - 1 \right) \right]^{1/2}
$$
$$
+ N^{-\frac{1}{2} + \eta_0} + \left( \frac{k}{n} \right)^{1/3} + \frac{\mathbb{E}|h - \theta|^3}{N^{1/2}(\mathbb{E}|h - \theta|^2)^{3/2}}
$$
$$
+ \left[ \frac{n}{N^{2\eta_0}} \frac{(1-p)\zeta_k}{k^2 \zeta_{1,\omega}} \right]^{\frac{1}{2}} \Bigg\}
$$

(7)

where $\tilde{C}$ is a positive constant. The proof follows the idea in (Peng et al., 2019) which decomposes the generalized incomplete U-statistic as a sum of complete U-statistic and a remainder. See section A.8 for details. Two asymptotic results can be induced from inequality equation 7:

1. If $0 < \alpha = \lim \alpha_n < \infty$ and $\frac{n}{N^{2\eta_0} k^2 \zeta_{1,\omega}} \to 0$, then $\frac{\sqrt{N}(U_{n,k,N;\omega} - \theta)}{\sqrt{k^2 \zeta_{1,\omega}/\alpha + \zeta_k}} \overset{d}{\to} N(0, 1)$ .

2. If $\lim \alpha_n = \infty$ and $\frac{n}{N^{2\eta_0} k^2 \zeta_{1,\omega}} \to 0$, then $\frac{\sqrt{N}(U_{n,k,N;\omega} - \theta)}{\sqrt{\zeta_k}} \overset{d}{\to} N(0, 1)$.

where we implicitly assume that $\lim_{n \to \infty} \zeta_k \ne \infty$. More generally, if $k/n \to 0, k^2/n \to \infty$ and $N \to \infty$, the asymptotic normality still holds under some conditions on moments of $h$. In summary, Theorem 3.2 generalizes the results in (Mentch & Hooker, 2016; Peng et al., 2019; Ghosal & Hooker, 2021) by directly assuming the resampling scheme is sampling without replacement and providing sharp Berry-Esseen bounds for asymptotic normality of RLF. Unlike the bounds in (Peng et al., 2019), inequality equation 7 provides one extra term which comes from the difference between uniformly sampling without replacement and Bernoulli sampling (See section A.8). It is worth mentioning that the asymptotic results in small-$N$ setting provide a theoretical support for people employing less number of base learners as long as the subsample size $k$ is appropriately designed.

## 3.3 COMPLEXITY ANALYSIS

The essential difference between RLF and RF is, a local RF is randomly determined to be fitted at certain nodes of each CART tree. Whether a local RF is fitted relies on a Bernoulli random variable. To analyze the computation time for fitting a RLF, we assume that a local RF will be fit at each non-terminal node of a RLT.

We borrow the notations used in Algorithm 1 and Algorithm 2. In the best case of balanced tree, the time flexibility of building a CART tree is $\mathcal{O}(m_{try} \cdot n \cdot \log_2 n)$ (Trevor Hastie & Friedman, 2009). If the optimal splitting leads to the extreme imbalanced cutting, the worst time flexibility would be $\mathcal{O}(m_{try} \cdot n^2)$ where the tree depth is $n$.

In the best case of building a Riemann-Lebesgue Tree, we have the following recursion relation

$$
T(n) = 2T(\frac{n}{2}) + \mathcal{O}(M_{local} \cdot m_{try} \cdot n \cdot \log_2 n)
$$

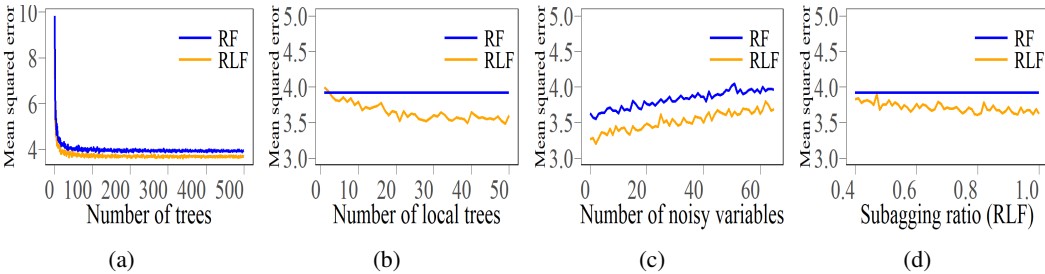

Figure 2: Test MSEs for RLF and RF. (a) Test MSE as a function of Number of trees, (b) Test MSE rate as a function of Number of local trees, (c) Test MSE as a function of Number of noisy variables, (d) Test MSE as a function of Subagging ratio.

where $T(\cdot)$ is a measurement of the runtime of a Riemann-Lebesgue Tree. By the limited fourth case in Master theorem (Cormen et al., 2009), $T(n) = \mathcal{O}(M_{local} \cdot m_{try} \cdot n \cdot \log_2^2 n)$. Then the best time complexity of RLF is $\mathcal{O}(M \cdot M_{local} \cdot m_{try} \cdot n \cdot \log_2^2 n)$, which has polylogarithmic runtime. With similar argument, we can see the worst time complexity of RLF is $\mathcal{O}(M \cdot M_{local} \cdot m_{try} \cdot n^2 \cdot \log_2 n)$.

In summary, we observe that the exact difference in complexity in best and worst comes from the factor $M_{local} \cdot \log_2 n$. This implies that the difference of complexity between classical RF and RLF increases when the size of dataset $n$ or the number of local trees $M_{local}$ increases. Table S4 in appendix compares the time efficiency of these two ensemble methods in many datasets. The results show that RLF is relatively slow but still comparable to RF when dataset is in small-to-medium scale. While in large datasets, RLF and RF both have low time efficiency. How to implement RLF in parallelization is one of our future directions.

## 4 EXPERIMENTS

### 4.1 SPARSE MODEL

To explore the performance of RLF in sparse model, we consider the following regression model in (Trevor Hastie & Friedman, 2009):

$$Y = 10 \cdot \prod_{j=1}^{5} e^{-2X_j^2} + \sum_{j=6}^{35} X_j + \varepsilon, \quad \varepsilon \sim N(0, \sigma^2)$$

where $\mathbf{X}_j$ is the $j$-th dimension of an observation $\mathbf{X}$. The response $\mathbf{Y} \in \mathbb{R}$ and random sample $X$ will be i.i.d and uniformly distributed on the 100-dimension unit cube $[0,1]^{100}$. In this case, the effective dimension is 35. And $\sigma$ is set to be 1.3 so that the signal-to-noise ratio is approximately 2. Fig. 2 summarizes Test MSEs for RLF and RF as functions of different hyperparameters. As we can see, given data-driven $\tilde{p}$, RLF outperforms RF in almost all experiment settings. See detailed experiment settings and analysis in section A.2 of appendix.

### 4.2 REAL DATA PERFORMANCE

We used 10-folds stratified cross-validation [2] to compare the performance of RLF and RF on 30 benchmark real datasets from (Fischer et al., 2023). The size of datasets ranges from five hundred to nearly fifty thousand observations. For time efficiency and consistency, we set the number of trees in RLF and RF to be 100 and subtrees in local RF in Lebesgue part of RLF to be 10, i.e $M_{local}$=10. See section A.1 for the detail of datasets and statistical significance tests employed.

We observe that RLF reaches roughly the same mean-squared-error as RF for most datasets and outperforms RF in 20 datasets where eight of them are statistically significant. The binomial test for win/loss ratio of RLF also shows that RLF does better job than RF statistically.

---

[2] That means stratified sampling method was employed to ensure the distribution of response is the same across the training and testing sets

Table 1: Mean squared error and corresponding 95% margin of error (based on 10-folds cross-validation) for Riemann–Lebesgue Forest(100,10) vs.Random Forest(100)

| Dataset | RLF(100,10) | RF(100) | Dataset | RLF(100,10) | RF(100) |
|---|---|---|---|---|---|
| FF | **4205.21 ±5612.77** | 4303.31±5432.01 | CPU | **5.72±0.63** | 5.92±0.59 |
| SP | 1.90±0.87 | **1.88±0.82** | KRA | **0.0188 ±0.00062**\* | 0.0214 ±0.00059 |
| EE | **1.258±0.32** | 1.262±0.35 | PUMA | **4.97e-4 ±1.38e-5**\* | 5.14e-4±1.67e-5 |
| CAR | **5374313 ±787019.9** | 5390741 ±840508.3 | GS | **1.2e-4 ±4.06e-6**\* | 1.5e-4±4.71e-6 |
| QSAR | **0.7568±0.14** | 0.7594±0.13 | BH | **0.01 ±0.01** | 0.011±0.012 |
| CCS | 28.34±4.87 | **28.19 ±4.09** | NPP | **5.97e-7±6.84e-8** | 6.5e-7±9.10e-8 |
| SOC | 587.87±205.53 | **565.08 ±204.29** | MH | **0.02±0.00093**\* | 0.022 ±0.0011 |
| GOM | **240.67±38.74** | 247.61±39.16 | FIFA | **0.5775±0.026** | 0.5796±0.024 |
| SF | **0.66±0.21** | 0.67 ±0.2 | KC | 0.041±0.0013 | **0.037±0.0013**\* |
| ASN | **11.87 ±1.28**\* | 13.02±1.30 | SUC | 83.16±2.26 | **82.38±2.27** |
| WINER | 0.3319±0.034 | **0.3299 ±0.035** | CH | **0.056±0.0026**\* | 0.059±0.0026 |
| AUV | **8073182±1513348**\* | 9368024±833367.9 | HI | 217.64±4.91 | **212.12±4.42**\* |
| SG | 0.01367±0.0045 | **0.01364±0.0046** | CPS | **0.2766±0.0077** | 0.2772±0.0068 |
| ABA | 4.63 ±0.54 | **4.58±0.56** | PP | **11.94±0.16**\* | 12.09±0.13 |
| WINEW | 0.3670±0.021 | **0.3612±0.022** | SA | **6.13±0.16** | 6.26±0.21 |

\*The better performing values are highlighted in bold and significant results are marked with "*".

### 4.3 Tuning of Splitting control probability $\tilde{p}$

Instead of using data-driven $\tilde{p}$, we can set the control probability $\tilde{p}$ as a tunable parameter when constructing a RLT. That is to say, we set $\tilde{p}$ be a fixed value for all nodes in a RLT. For instance, under the same sparse model defined in section 4.1, Fig.3(a) indicates that the sparse model favors more Lebesgue cuttings.[3] This is consistent with the intuition that noisy variables weaken the effectiveness of Riemann cutting. In this section, we provide two one-dimensional examples to illustrate the flexibility of RLF with tuned $\tilde{p}$. One is designed to have small signal-to-noise ratio. The other is a mixture model with prior probability where we anticipate that RLF will perform better because of nature the Lebesgue type cutting. Test MSEs of RLF as functions of $\tilde{p}$ in Fig S4(a) and S4(b) exhibit the potential benefit of tuning control probability $\tilde{p}$. See details of tuning procedure in section A.9.

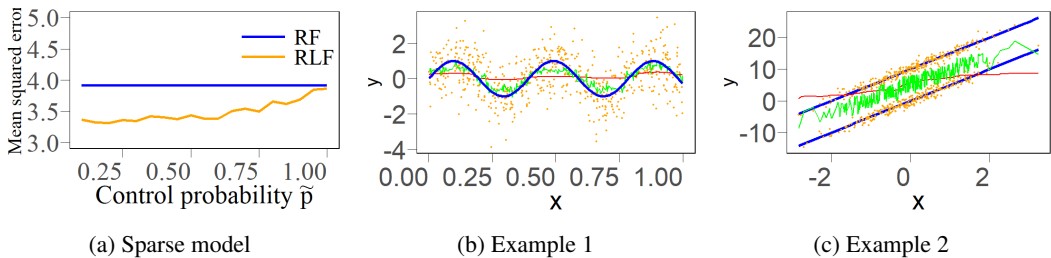

|(a) Sparse model|(b) Example 1|(c) Example 2|

Figure 3: (a) Test MSEs for RLF and RF as functions of control probability $\tilde{p}$. Orange points in (b) and (c) represent test samples generated by two models. Blue solid lines are the underlying functions ;Green lines are the predicted curves for optimal RLFs while the red lines are the predicted curve of optimal RF in two examples.

#### Example 1: Small signal-to-noise ratio

Figure 3(b) demonstrates the predicted regression curves from tuned RLF and tuned RF in one-dimensional case. Our synthetic model is based on the sine function used in (Cai et al., 2020) and (Cai et al., 2023):

$$Y = \sin(16X) + \varepsilon, \quad \varepsilon \sim N(0, \sigma^2)$$

where $X \sim \text{Unif}[0, 1]$ and $\sigma = 1$. Predicted curves in Fig.3(b) show that tuned RLF is robust to relative high level of noise and it is capable to identify the pattern in response while traditional RF is too conservative in this case. Table S5 (See appendix) lists validation MSE of top 10 models with tuning parameters for RF and RLF. Based on the top 10 models, we can see RLF generally

---

[3]Since RF doesn't have parameter $\tilde{p}$ so the Test MSE curve of RF is a constant function w.r.t $\tilde{p}$

outperforms RF in this case. Indeed, the testing MSE under best RF is 1.283 while the testing MSE of best RLF is 1.018 (See Table S6 in Appendix). Another interesting phenomenon is , when the noise level is high, RLF prefers using more Riemann type cuttings to achieve better performance as top 5 tuned RLFs in Table S6 have relatively large value of $\tilde{p}$.

EXAMPLE 2: MIXTURE MODEL WITH PRIOR PROBABILITY

In this section, we consider a mixture model as follows:

$$Y = \begin{cases} 5X + \varepsilon & \text{if } C = 1 \\ 10 + 5X + \varepsilon & \text{if } C = 2 \end{cases} \tag{8}$$

where $X, \varepsilon \overset{i.i.d}{\sim} N(0, 1)$ and the random variable $C$ is uniformly distributed on $\{1, 2\}$. To generate an observation of response, we first randomly pick a value of state variable $C$ from $\{1, 2\}$ and generate $Y$ according to the model defined in equation 8. In this way, $Y$ will follow a mixture Gaussian distribution. Figure 3(c) demonstrates that RLF is able to capture complicated distribution of response with the help of Lebesgue cuttings while traditional RF fails in detecting the complex pattern in response.

Similarly, Table S7 (See appendix) lists validation MSEs of top 10 models with tuning parameters and the testing error of best RLF (29.87) is smaller than that of best RF (35.17) showing that RLF also beats RF in this case. It is worth mentioning that Table S7 gives us a clue that when response $Y$ has mixture distribution, RLF favors more Lebesgue cuttings since top 5 tuned RLF models tend to employ relatively small value of $\tilde{p}$.

## 4.4 EXTRA EXPERIMENTS OF RLF WITH TUNED $\tilde{p}$

We perform extra experiments on 26 datasets listed in Table 1 to show the strength of RLF after parameter tuning. Four large datasets are excluded due to the time efficiency of parameter tuning. Table 2 shows that tuned RLF still outperforms tuned RF in many real world datasets and wins more often than in original experiments. More specifically, tuned RLF wins in 23 datasets among which 12 of them are statistically significant. See A.10 for detailed settings for tuning process.

Table 2: Averaged MSEs and 95% marginal errors for Best RLF and Best RF

| Dataset | Best RLF | Best RF | Dataset | Best RLF | Best RF |
|---|---|---|---|---|---|
| FF | **4221.49 ±9191.91** | 4376.29±8922.14 | ABA | 4.66±0.39 | **4.63±0.38** |
| SP | **1.78 ±0.78** | 1.98±0.73 | WINEW | **0.398±0.025** | 0.403±0.0264 |
| EE | **0.54 ±0.29*** | 1.50±0.35 | CPU | **5.67±0.58*** | 6.15±0.43 |
| CAR | **5194266 ±1287865** | 5622327±1482889 | KRA | **0.019±0.0008*** | 0.024±0.0013 |
| QSAR | **0.79 ±0.10** | 0.80±0.13 | PUMA | **0.00048±1.35e-05*** | 0.00053±1.82e-05 |
| CCS | **27.11 ±3.52*** | 37.23±7.0 | GS | **0.00011±7.95e-06*** | 0.00016±8.07e-06 |
| SOC | **405.63±376.21** | 621.75±528.39 | BH | **0.0061±0.0032** | 0.0069±0.0040 |
| GOM | **255.49±56.60** | 263.16±64.10 | NPP | **7.79e-07±1.96e-07*** | 1.08e-06±1.95e-07 |
| SF | 0.67±0.19 | **0.62±0.21** | MH | **0.021±0.0017*** | 0.023±0.0012 |
| ASN | **5.67±1.07*** | 14.14±2.00 | FIFA | **0.578±0.011** | 0.581±0.010 |
| WINER | **0.347±0.026** | 0.349±0.026 | SUC | **89.67±11.40** | 91.80±10.36 |
| AUV | **4192163±1236743*** | 9878887±1670280 | CH | **0.056±0.018*** | 0.062±0.019 |
| SG | **0.0137±0.0049*** | 0.0143±0.0049 | CPS | 0.279±0.0061 | **0.27±0.0060** |

The better performing values are highlighted in bold and significant results are marked with "*".

## 5 DISCUSSION AND LIMITATION

Theorem 3.1 has shown the benefit of Lebesgue cutting in reducing the $L_2$ error of RLF. Section 4.3 further demonstrates the flexibility of RLF in many complicated models and real world datasets with tuned $\tilde{p}$. The asymptotic normality in section 3 is useful for statistical inference such as confidence intervals and hypothesis testings (Mentch & Hooker, 2016). It's possible to obtain a Berry-Esseen bound of RLF in large $N$-setting by similar arguments. However, this bound can be worthless in practice as it's requires $N \gg n$ and we decide not to pursue it in this paper.

Although the experiment results show that tuned $\tilde{p}$ is superior than data-driven $\tilde{p}$, the cross-validation tuning process is less time efficient than data-driven methods in large datasets. How to choose the optimal value of $\tilde{p}$ is an interesting problem. On the other hand, we employ a Bernoulli random variable to determine the splitting type at each non-terminal node. It's possible that there are other better regularization methods to control the overfitting resulted from Lebesgue splitting. Connecting RLT with boosting would be another riveting direction as RLT is essentially a new kind of base learner in ensemble methods.

As we discussed in section 3.3, time efficiency is the main limitation of RLF. For prediction part, we employ random forest locally which is powerful but time consuming for large dataset. Developing a more efficient local model for RLF would benefit RLF in large datasets. The time efficiency of current RLF can be improved by employing less number of subtrees in local forest and setting larger value of control probability $\tilde{p}$ so that each RLT won't perform too many Lebesgue splittings.

Readers may view the use of Riemann-Lebesgue Tree as counterintuitive if the local models (e.g local forests) can estimate response Y accurately. However, the local models don't have to be as good as the RLF or RLT since we only require the local models to indicate the direction, not the precise prediction of new points. What's more, when the response is complicated, as indicated in example 2 (mixture model with prior probability), we can employ more Lebesgue type splittings in RLF to partition the range of response $Y$. As a result, the local distribution of $Y$ becomes simpler which will relax the requirement of the precision of local models. The flexibility of RLF comes from the control probability $\tilde{p}$, which controls the number of Lebesgue type splittings in each RLT and can be tailored to different cases.

In more general scenarios, how to cope with possible missing values deserves deep inspection since RLF replies heavily on the information from all variables and it's possible that the prediction of local RF in RLF could be misled by missing values. Actually, in practice, we can perform imputation by rough average/mode, or by an averaging/mode based on proximities during the data preprocessing, which is out of the scope of our current paper.

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

## A    APPENDIX

### A.1    DATA PREPARATION AND STATISTIC TOOLS USED IN REAL DATA EXPERIMENTS

Table S3  summarizes the datasets we used in experiments. Note that there are 35 datasets in original benchmarks. We didn't perform comparison on the datasets with the number of instances more than 50,000 since the complexity of RLF is relatively higher than RF. We also excluded datasets with missing values whose results might be unfair. We took logarithm transformation for some datasets to alleviate the impact of skewness. As described in Algorithm 2, we set the subagging ratio to be 0.632, which should have the same efficiency with bootstrapping (Efron & Tibshirani, 1997)

Table S3 : Real datasets used for the experiments,sorted by size

| Dataset | Observations | Numerical features | Symbolic features | Log transformation |
|---|---|---|---|---|
| Forestfire (FF) | 517 | 9 | 2 | No |
| Student performance (SP) | 649 | 14 | 17 | No |
| Energy efficiency (EE) | 768 | 9 | 0 | No |
| Cars(CAR) | 804 | 18 | 0 | No |
| QSAR fish toxicity (QSAR) | 908 | 7 | 0 | No |
| Concrete Compressive Strength (CCS) | 1030 | 9 | 0 | No |
| Socmob(SOC) | 1056 | 2 | 4 | No |
| Geographical Origin Of Music (GOM) | 1059 | 117 | 0 | No |
| Solar Flare (SF) | 1066 | 3 | 8 | No |
| Airfoil Self-Noise (ASN) | 1503 | 6 | 0 | No |
| Red wine quality (WINER) | 1599 | 12 | 0 | No |
| Auction Verification (AUV) | 2043 | 7 | 1 | No |
| Space Ga (SG) | 3107 | 7 | 0 | No |
| Abalone (ABA) | 4177 | 8 | 1 | No |
| Winequality-white (WINEW) | 4898 | 12 | 0 | No |
| CPU Activity (CPU) | 8192 | 22 | 0 | No |
| Kinematics of Robot Arm (KRA) | 8192 | 9 | 0 | No |
| Pumadyn32nh (PUMA) | 8192 | 33 | 0 | No |
| Grid Stability (GS) | 10000 | 13 | 0 | No |
| Brazil Housing (BH) | 10692 | 6 | 4 | Yes |
| Naval propulsion plant (NPP) | 11934 | 15 | 0 | No |
| Miami housing (MH) | 13932 | 16 | 0 | Yes |
| Fifa (FIFA) | 19178 | 28 | 1 | Yes |
| Kings county (KC) | 21613 | 18 | 4 | Yes |
| Superconductivity (SUC) | 20263 | 82 | 0 | No |
| Califonia housing (CH) | 20460 | 9 | 0 | Yes |
| Health insurance (HI) | 22272 | 5 | 7 | No |
| Cps88wages (CPS) | 28155 | 3 | 4 | Yes |
| Physiochemical Protein (PP) | 45730 | 10 | 0 | No |
| Sarcos (SA) | 48933 | 22 | 0 | No |

We employed a corrected resampled t-test (Nadeau & Bengio, 2003; Landwehr et al., 2003), to identify whether one method significantly outperforms another at $5\%$ significance level. This test rectifies the dependencies of results induced by overlapped data points and has correct size and good power. The corresponding statistic is

$$t = \frac{\frac{1}{V} \sum_{t=1}^{V} r_t}{\sqrt{(\frac{1}{V} + \frac{n_2}{n_1})\hat{\sigma}^2}}$$

where $V$ is the number of validation experiments performed (ten in our case), $r_t$ is the difference in MSE between RLF and RF on $t$-th fold. $\hat{\sigma}$ is the sample standard deviation of differences. $n_1$ is the number ofdata points used for training and $n_2$is the number of testing cases. In our case, $n_2/n_1 = 1/9$ since we used 10-folds cross-validation.

## A.2 EXPERIMENT DETAILS IN SPARSE MODEL

There are 1000 training cases and 500 test observations. Each subplot in Figure 2 illustrates the curves of mean squared error for RLF and RF, as functions of a single hyperparameter. The default setting for RLF is $M = 100, M_{local} = 10, \alpha = 0.632, M_{node} = 5$. Same setting except $M_{local}$ applies to RF. To obtain, for example the test MSE as a function of number of global trees, we set $M = 1, 2, ..., 500$ while other parameters remain the same with default setting. Similar strategy was applied to the other three hyperparameters we are interested.

In Fig.2(a) , we let RLF and RF share the same the number of trees. We observe that the test MSE for RLF decreases more than that of RF as the number of trees increases. In Fig.2(b) , RF is fitted under default setting while we only changed the number of local trees used in local random forest of RLF. As we can see, large number of local trees indeed benefits the performance of RLF. On the other hand, in order to control the computation time burden, the number of local trees should not be too large. Fig.2(c)  implies that increasing the number of noisy variables in the regression model will impair both the performance of RLF and RF. Nevertheless, RLF always outperforms RF, which is anticipated. To see the impact of subagging ratio on RLF, we first controlled the result from ordinary RF under default setting and increased subagging ratio, which determines the subsample size in RLF, from 0.4 to 1. The test MSE curve for RLF in Fig.2(d)  indicates that even a small subagging ratio could generatea decent result. We can therefore set the subagging ratio relatively small to make RLF efficient.

## A.3 ALGORITHM FOR RLF

---

**Algorithm 2** Riemann-Lebesgue Forest prediction at $\mathbf{x}$

---

**Require:** Original Training data $\mathcal{D}_n = \{(\mathbf{X}_i, Y_i), i = 1, ..., n\}$ with $\mathbf{X}_i \in [0, 1]^d$. Minimum node size $M_{node}, m_{try} \in \{1, 2, ..., d\}$. Number of trees $M > 0$, Number of trees used in local random forests $M_{local} > 10$ , $k \in \{1, 2, ..., n\}$ and $\mathbf{x}$.
  1: **for** $i = 1$ to $M$ **do**
  2:     Select $k$ points without replacement from $\mathcal{D}_n$.
  3:     Use selected $k$ points to fit a Riemann-Lebesgue Tree described in Algorithm 1. Hyperparameters such as $M_{node}, m_{try}$ of RLF are shared with each Riemann-Lebesgue Tree.
  4:     Compute $T_{\mathbf{x}}^{(\omega_i)}((X_{i1}, Y_{i1}), ..., (X_{ik}, Y_k))$ , the predicted value of the Riemann-Lebesgue Tree at $\mathbf{x}$, which is the average of $Y_i's$ falling in the terminal node of $\mathbf{x}$ in.
      **return** Compute the random forest estimate $R_n(\mathbf{x})$, which is the final predicted value of $\mathbf{x}$ from Riemann-Lebesgue Forest.

---

## A.4 INCOMPLETE U-STATISTICS

Before give the definition of incomplete U-statistics, we first give a brief introduction of U-statistics. We borrow notations from (Vaart, 2000). Suppose we have i.i.d samples $X_1, ..., X_n$ from an unknown distribution, and we want to estimate a parameter $\theta$ defined as follows

$$\theta = \mathbb{E}h(X_1, ..., X_r)$$

where the function $h$ is permutation symmetric in its r arguments and we call $h$ as a kernel function with order $r$. Then a U-statistic with kernel $h$ is defined as

$$U = \frac{1}{\binom{n}{r}} \sum_{\beta} h(X_{\beta_1}, ..., X_{\beta_r})$$

where the sum is taken over all all unordered subsets $\beta$ with $r$ distinct elements from $\{1, ..., n\}$ .

Note that the $U$-statistic is an unbiased estimator for $\theta$ and has smaller variance than a single kernel $h$. In practice, it's unrealistic to average all $\binom{n}{r}$ kernels (trees). To make $U$-statistics more useful in applications, we can define an incomplete $U$-statistic which utilizes smaller number of subsets as follows:

$$U_{n,r,N} = \frac{1}{N} \sum_{i=1}^{N} h(X_{\beta_1,...,X_{\beta_r}})$$

when $N = \binom{n}{r}$, we obtain a complete $U$-statistic.

When the total number of subsamples of size $r$ taken from a sample of size $n$ is large, it may be convenient to use instead an 'incomplete' U-statistic based on $N$ suitably selected subsamples. Asymptotically, it can be shown that an incomplete $U$-statistic may be asymptotically efficient compared with the 'complete' one even when $N$ increases much less rapidly than the total number. See more explanations and applications for complete/incomplete $U$-statistics in (Vaart, 2000; Lee, 1990; Peng et al., 2019; Chen et al., 2010). In a Riemann-Lebesgue Forest, each tree can be viewed as a random kernel since it is permutation symmetric w.r.t inputs. The randomness of the kernel comes from feature bagging and subsampling and cutting types. Similar definition can be applied to RF as well.

## A.5 CONSISTENCY OF RLF

Based on the results from (Scornet et al., 2014), it's relatively easy to derive the consistency of RLF under an additive regression model as follows:

$$Y = \sum_{j=1}^{p} m_j(\mathbf{X}_j) + \varepsilon \tag{9}$$

where $\mathbf{X} = (\mathbf{X}_1, .., \mathbf{X}_p)$ is uniformly distributed over $[0,1]^p$, the noise $\varepsilon \sim N(0, \sigma^2)$ has finite variance and is independent of $\mathbf{X}$. Each component function $m_j$ is continuous.

When there are only $S(< p)$ effective dimensions of the model, the sparse version of equation 9 becomes

$$Y = \sum_{j=1}^{S} m_j(\mathbf{X}_j) + \varepsilon$$

which is exactly the condition where RLF may perform better. Therefore the additive regression model is a good framework for studying the consistency of RLF. Since the Lebesgue part of each Riemann-Lebesgue Tree is essentially splitting the response $Y$ with CART-criterion, many consistency results for CART can be applied to RLT directly.

For example, Breiman et al. (1984) proved the consistency of CART under the assumptions of shrinking diameter of cell partition and lower bounds of empirical distribution of $X$. Scornet et al. (2014) shown the consistency of RF under the assumption of additive regression model and appropriate complexity of the tree partition.

We now state one version of consistency adapted to RLF where we assume that the total number of nodes $t_n$ in each RLT approaches to infinity more slowly than the subsample size $k_n$. The proof follows immediately from the argument for Theorem 1 in (Scornet et al., 2014).

**Theorem A.1.** *Assume the response $Y$ is generated from the sparse model defined in equation 9. Then, given $k_n \to \infty, t_n \to \infty$ and $t_n(log k_n)^9/k_n \to 0$ where $k_n$ is the subagging size and $t_n$ is the number of terminal nodes in each Riemann-Lebesgue Tree, we have*

$$\lim_{n \to \infty} \mathbb{E}_X[h^*_{n,k,N}(\mathbf{X}) - m(\mathbf{X})]^2 = 0$$

*where $m(\mathbf{X}) = \mathbb{E}[Y|X]$, $h^*_{n,k,N}(\mathbf{X}) = \mathbb{E}_{\omega,Z}[h^{(\omega)}(X; Z)]$. The consistency of empirical averaged RLF estimate follows from the law of large numbers.*

Note that Theorem A.1 still holds even when we select all data points for each tree, which means controlling the number of nodes $t_n$ via minimal size of nodes $M_{node}$ is sufficient for the error bound of RLF. According to (Scornet et al., 2014), the term $(\log k_n)^9$ results from the assumption of Gaussian noise. We can replace it by a bounded random variable so that the term $(\log k_n)^9$ becomes $\log k_n$, which can be regarded as the complexity of a single RLT partition,

## A.6 RUNNING TIME FOR RLF AND RF

For completeness, Table S4 compares averaged running time (in seconds) of RLF,RF on our selected datasets in manuscript. For RLF, $M = 100, M_{local} = 10$ and $\tilde{p} = 0.8$. For RF, $M = 100$. Other parameters are set by default values. We ran 10-fold cross-validation. We calculated the running time as sum of training time and prediction time.

Table S4 : Average running time (in seconds) for RLF(100),RF(100)

| Dataset | Observations | # Features | RLF | RF |
|---|---|---|---|---|
| FF | 517 | 11 | 1.28±0.97 | 0.16± 0.55 |
| SP | 649 | 31 | 1.73±0.1 | 0.32±0.04 |
| EE | 768 | 9 | 0.54±0.012 | 0.052±0.0036 |
| CAR | 804 | 18 | 0.86±0.07 | 0.084±0.0035 |
| QSAR | 908 | 7 | 1.37±0.067 | 0.14±0.035 |
| CCS | 1030 | 9 | 1.53±0.079 | 0.16±0.04 |
| SOC | 1056 | 6 | 0.72±0.052 | 0.073±0.0038 |
| GOM | 1059 | 117 | 15.72±0.90 | 2.37±0.074 |
| SF | 1066 | 11 | 0.90±0.028 | 0.12±0.004 |
| ASN | 1503 | 6 | 1.15±0.14 | 0.11±0.004 |
| WINER | 1599 | 12 | 3.13±0.15 | 0.38±0.0042 |
| AUV | 2043 | 8 | 1.89±0.12 | 0.22±0.031 |
| SG | 3107 | 7 | 7.94±0.47 | 1.02±0.044 |
| ABA | 4177 | 9 | 13.57±0.68 | 2.08±0.10 |
| WINEW | 4898 | 12 | 15.73±0.82 | 4.10±0.13 |
| CPU | 8192 | 22 | 68.76±3.07 | 16.39±0.84 |
| KRA | 8192 | 9 | 32.66±1.59 | 8.45±0.56 |
| PUMA | 8192 | 33 | 112.14±3.71 | 24.36±1.06 |
| GS | 10000 | 13 | 70.48±5.8 | 22.53±3.00 |
| BH | 10692 | 10 | 123.09±5.38 | 31.48±01.15 |
| NPP | 11934 | 15 | 180.33±33.76 | 39.55±8.43 |
| MH | 13932 | 16 | 150.73±5.23 | 55.12±4.28 |
| FIFA | 19178 | 29 | 550.83±57.34 | 388.54±75.08 |
| SUC | 20263 | 82 | 1122.48±125.88 | 993.17±102.78 |
| CH | 20460 | 9 | 714.86±79.44 | 340.15±43.62 |
| KC | 21613 | 22 | 1510.17±65.83 | 651.06±15.03 |
| HI | 22272 | 12 | 1370.773±163.53 | 486.55±43.02 |
| CPS | 28155 | 7 | 206.32±41.58 | 58.75±12.78 |
| PP | 45730 | 10 | 2551.45±398.63 | 2149.19±260.17 |
| SA | 48933 | 22 | 4505.45±1098.13 | 2347.74±258.91 |

## A.7 PROOF OF THEOREM 3.1

For simplicity, we assume one-dimension case, i.e $Y = f(X) + \varepsilon$. Let $A_1, A_2$ be a nontrivial partition of the initial outcome space, by the law of total variance or variance decomposition formula, we have

$$
\begin{aligned}
Var(Y) = {} & Var(Y|A_1)P(A_1) + Var(Y|A_2)P(A_2) \\
& + \mathbf{E}[Y|A_1]^2(1 - P(A_1))P(A_1) + \mathbf{E}[Y|A_2]^2(1 - P(A_2))P(A_2) \\
& - 2\mathbf{E}[Y|A_1]P(A_1)\mathbf{E}[Y|A_2]P(A_2)
\end{aligned}
$$

In Lebesgue cutting, we have $A_1 = \{Y > a\}, A_2 = \{Y \le a\}$. Then the theoretical variance reduction of $Y$ in the initial outcome space can be written as following:

$$
\begin{aligned}
Var(Y) - Var(Y|A_1)P(A_1) - Var(Y|A_2)P(A_2) = {} & \mathbf{E}[Y|Y > a]^2(1 - P(Y > a))P(Y > a) \\
& + \mathbf{E}[Y|Y \le a]^2(1 - P(Y \le a))P(Y \le a) \\
& - 2\mathbf{E}[Y|Y > a]P(Y > a)\mathbf{E}[Y|Y \le a]P(Y \le a)
\end{aligned}
$$

To find optimal splitting point $a$ which gives the maximal variance reduction, we should solve the following optimization problem.

$$\max_a \left[ \left( (\mathbf{E}[Y|Y > a] - \mathbf{E}[Y|Y \leq a])^2 \cdot (1 - P(Y > a))P(Y > a) \right) \right]$$

$$= \max_a \left[ \left( \left( P(Y \leq a)\mathbf{E}[Y\mathbf{1}_{Y>a}] - P(Y > a)\mathbf{E}[Y\mathbf{1}_{Y\leq a}] \right)^2 \cdot \frac{1}{(1 - P(Y > a))P(Y > a)} \right) \right]$$

$$= \max_a \left[ \frac{\mathbf{E}[Y\mathbf{1}_{Y>a}]^2}{(1 - P(Y > a))P(Y > a)} \right] \quad \text{(WLOG, we assume } \mathbf{E}[Y] = 0)$$

$$= \max_a \left[ L(a) \right]$$

where

$$L(a) = \frac{\mathbf{E}[Y\mathbf{1}_{Y>a}]^2}{(1 - P(Y > a))P(Y > a)}$$

Similar argument applies for Riemann cutting. Let Riemann partition be $B_1 = \{X > b\}, B_2 = \{X \leq b\}$, then the corresponding optimization problem for Riemann cutting would be

$$\max_b \left[ \frac{\mathbf{E}[Y\mathbf{1}_{X>b}]^2}{(1 - P(X > b))P(X > b)} \right] = \max_b \left[ R(b) \right]$$

where

$$R(b) = \frac{\mathbf{E}[Y\mathbf{1}_{X>b}]^2}{(1 - P(X > b))P(X > b)}$$

To maximize function $L(a)$, $a$ will go through all possible values of $Y$ which essentially considers all possible partitions w.r.t $Y$. While for $R(b)$, when we go through all possible values of $X$, it does not necessary check all possible cutting w.r.t $Y$. Therefore, the numerator in $L(a)$ has more possible values. On the other hand, denominators in $L(a)$ and $R(b)$ have the same range from $0$ to $1$.

These two observations tell us that function $L(a)$ has larger range set. We can now conclude that

$$\max_a \left[ \frac{\mathbf{E}[Y\mathbf{1}_{Y>a}]^2}{(1 - P(Y > a))P(Y > a)} \right] \geq \max_b \left[ \frac{\mathbf{E}[Y\mathbf{1}_{X>b}]^2}{(1 - P(X > b))P(X > b)} \right] \tag{10}$$

In other words, the optimal Lebesgue cutting can reduce more variance than optimal Riemann cutting (CART) does. We can also see the noise doesn't affect the conclusion since $Y$ already absorbed noises in practice.

**Another analysis of equation 10 from discrete case:**

In discrete case, we need to compare $L(j^*, z^*)$ and $\tilde{L}(z_L^*)$ as defined in equation 4 and equation 5.

Note that the set of possible partitions of current set of responses induced by the Riemann cutting of covariate $\mathbf{X}$ is a subset of that of Lebesgue cutting whose feasible set is essentially induced by response directly. Note that the objective functions $L(j, z)$ and $\tilde{L}(z)$ have the same form, the one with larger feasible set should have larger optimal value.

Suppose we have $n$ points $\{(X_1, Y_1), ..., (X_n, Y_n)\}$ whose y values are all distinct in a certain non-terminal node and both cutting types employ CART split criterion, then Lebesgue cutting will go through all $(n + 1)$ splitting points in $Y$ directly, which essentially gives the largest feasible set of optimization problem. On the other hand, in Riemann type cutting, the space of $Y$ is indirectly partitioned by going through all possible splitting points in direction $X^{(j)}$. It's not necessary that this procedure will take care all $(n + 1)$ splitting points in $Y$. For example, when $X^{(j)}$ is a direction with only three distinct values , $z^{(j)}$ can only have four choices of splitting point which will restrict the possible values of $\bar{Y}_{A_L}, \bar{Y}_{A_R}$ in optimizing $L(j, z)$ and leads to a smaller feasible set.

### A.8    PROOF OF THEOREM 3.2

We basically follows the idea in (Peng et al., 2019) which decomposes the generalized incomplete U-statistic as a sum of complete U-statistic and a remainder. To deal with the random kernel, we utilize the conclusions based on extended Hoeffding decomposition (Peng et al., 2019). For the simplicity of notation, it's harmless to assume that $\theta = 0$.

For $0 < \eta_0 < \frac{1}{2}$, we first decompose $U_{n,k,N,\omega}/\sqrt{k^2\zeta_{1,\omega}/n + \zeta_k/N^{2\eta_0}}$ as follows:

$$
\frac{U_{n,k,N,\omega}}{\sqrt{\frac{k^2\zeta_{1,\omega}}{n} + \frac{\zeta_k}{N}}} = \frac{\frac{1}{\binom{n}{k}}\sum_{(n,k)}\frac{\rho_i}{p}h^{(\omega_i)}(Z_{i_1},...,Z_{i_k}) + \frac{1}{\binom{n}{k}}\sum_{(n,k)}\frac{W_i - \rho_i}{p}h^{(\omega_i)}(Z_{i_1},...,Z_{i_k})}{\sqrt{\frac{k^2\zeta_{1,\omega}}{n} + \frac{\zeta_k}{N}}}
$$
$$
= \frac{A + \frac{1}{\binom{n}{k}}\sum_{(n,k)}\frac{W_i - \rho_i}{p}h^{(\omega_i)}(Z_{i_1},...,Z_{i_k})}{\sqrt{\frac{k^2\zeta_{1,\omega}}{n} + \frac{\zeta_k}{N}}}
\tag{11}
$$

where

$$
A = \frac{1}{\binom{n}{k}}\sum_{(n,k)}\frac{\rho_i}{p}h^{(\omega_i)}(Z_{i_1},...,Z_{i_k})
$$
$$
= \frac{\hat{N}}{N}\frac{1}{\hat{N}}\sum_{(n,k)}\rho_i h^{(\omega_i)}(Z_{i_1},...,Z_{i_k})
\tag{12}
$$

and $\rho_i \overset{i.i.d}{\sim} Bernoulli(p)$, $p = N/\binom{n}{k}$ and $\hat{N} = \sum_{(n,k)}\rho_i$. We can see $E[\hat{N}] = N$. WLOG, we can assume $\theta_{n,k,N} = 0$. Then we have

$$
A = \frac{\hat{N}}{N}B
\tag{13}
$$

where

$$
B = \frac{1}{\hat{N}}\sum_{(n,k)}\rho_i h^{(\omega_i)}(Z_{i_1},...,Z_{i_k})
$$

and $B$ is a complete generalized U-statistic with Bernoulli sampling as described in (Peng et al., 2019). According to Theorem 4 in Peng et al. (2019), we have

$$
sup_{z\in R}\left|P\left(\frac{B}{\sqrt{k^2\zeta_{1,\omega}/n + \zeta_k/N}}\right) - \Phi(z)\right| \leq C\left\{\frac{\mathbb{E}|g|^3}{n^{1/2}\zeta_{1,\omega}^{3/2}} + \frac{\mathbb{E}|h|^3}{N^{1/2}(\mathbb{E}|h|^2)^{3/2}}\right.
$$
$$
\left. + \left[\frac{k}{n}\left(\frac{\zeta_k}{k\zeta_{1,\omega}} - 1\right)\right]^{1/2} + \left(\frac{k}{n}\right)^{1/3} + N^{-\frac{1}{2}+\eta_0}\right\}
$$
$$
:= \varepsilon_0
\tag{14}
$$

where $0 < \eta_0 < 1/2$.

By definition, we have $\zeta_k = var(h) = \mathbb{E}|h|^2 = \Sigma_{k,n,\omega,\mathbf{Z}}^2$

Denote

$$
T = \frac{A}{\sqrt{k^2\zeta_{1,\omega}/n + \zeta_k/N}} = W + \Delta
$$

where

$$
W = \frac{B}{\sqrt{k^2\zeta_{1,\omega}/n + \zeta_k/N}}, \quad \Delta = (\frac{\hat{N}}{N} - 1)W
$$

According to the argument in proof of Theorem 4 in (Peng et al., 2019), it's easy to verify that

$$-P\left(z - |\Delta| \le W \le z\right) \le P\left(U \le z\right) - P\left(W \le z\right) \le P\left(z \le W \le z + |\Delta|\right)$$

Thus we only need to consider bounding $P(z \le W \le Z + |\Delta|)$. By Bernstein's inequality, we have

$$P\left(|\frac{\hat{N}}{N} - 1| \ge \varepsilon\right) \le 2exp\left(-\frac{-\varepsilon^2 N}{1 - p + \varepsilon/3}\right)$$

Let $\varepsilon = N^{-\beta}$ and note that $P(|Z| \ge N^\alpha) \le 2exp(-N^{2\alpha}/2)$, we have

$$
\begin{aligned}
P(|\Delta| \ge N^{-\beta+\alpha}) &\le P\left(|\frac{\hat{N}}{N} - 1| \ge N^{-\beta}\right) + P(|W| \ge N^\alpha) \\
&\le 2\exp\left(-\frac{N^{1-2\beta}}{1 - p + N^{-\beta}/3}\right) + P(|Z| \ge N^\alpha) + 2\varepsilon_0 \\
&\le 2\exp\left(-\frac{1}{(1-p)N^{2\beta-1} + N^{\beta-1}/3}\right) + 2\exp(-N^{2\alpha/2}) + 2\varepsilon_0 \\
&:= \varepsilon_1 + 2\varepsilon_0
\end{aligned}
\tag{15}
$$

Eventually, we can bound $P(z \le W \le z + |\Delta|)$ as follows:

$$
\begin{aligned}
P(z \le W \le z + |\Delta|) &\le P(z \le W \le z + |\Delta|, |\Delta| \le N^{-\beta+\alpha}) + P(|\Delta| \ge N^{-\beta+\alpha}) \\
&\le P(z \le W \le z + N^{-\beta+\alpha}) + \varepsilon_1 + 2\varepsilon_0 \\
&\le 2\varepsilon_0 + P(z \le Z \le z + N^{-\beta+\alpha}) + \varepsilon_1 + 2\varepsilon_0 \\
&\le 4\varepsilon_0 + \varepsilon_1 + \frac{1}{\sqrt{2\pi}}N^{-\beta+\alpha} \\
&:= 4\varepsilon_0 + \varepsilon_1 + \varepsilon_2
\end{aligned}
\tag{16}
$$

Let $\beta = 0.5 + \eta_1$ and $\alpha = \eta_1$, where $\eta_1 > 0$. It's easy to see $\varepsilon_1 \ll \varepsilon_2$ when $N$ is large and therefore

$$
\begin{aligned}
\sup_{z \in R}\left|P\left(\frac{A}{\sqrt{k^2\zeta_{1,\omega}/n + \zeta_k/N}}\right) - \Phi(z)\right| &= \sup_{z \in R}\left|P(T \le z) - \Phi(z)\right| \\
&\le \sup_{z \in R}\left|P(W \le z) - \Phi(z)\right| + \sup_{z \in R}\left|P(T \le z) - P(W \le z)\right| \\
&\le 5\varepsilon_0 + \varepsilon_1 + \varepsilon_2 \\
&= C\left\{\frac{\mathbb{E}|g|^3}{n^{1/2}\zeta_{1,\omega}^{3/2}} + \frac{\mathbb{E}|h|^3}{N^{1/2}(\mathbb{E}|h|^2)^{3/2}}\right. \\
&\quad + \left.\left[\frac{k}{n}\left(\frac{\zeta_k}{k\zeta_{1,\omega}} - 1\right)\right]^{1/2} + \left(\frac{k}{n}\right)^{1/3} + N^{-\frac{1}{2}+\eta_0} + N^{-\frac{1}{2}}\right\} \\
&:= \varepsilon_3
\end{aligned}
\tag{17}
$$

We observe that the factor of $\hat{N}/N$ only produces the extra term $N^{-\frac{1}{2}}$ in the final bound, which is similar to (Peng et al., 2019). We employ this technique one more time to achieve the bound for $U_{n,k,N,\omega}$.

Denote

$$\frac{U_{n,k,N,\omega}}{\sqrt{\frac{k^2\zeta_{1,\omega}}{n} + \frac{\zeta_k}{N}}} = \frac{A}{\sqrt{\frac{k^2\zeta_{1,\omega}}{n} + \frac{\zeta_k}{N}}} + C = \frac{A}{\sqrt{\frac{k^2\zeta_{1,\omega}}{n} + \frac{\zeta_k}{N}}} + \frac{D}{\sqrt{k^2\zeta_{1,\omega}/n + \zeta_k/N}}$$

where A is defined as above and

$$C = \frac{\frac{1}{\binom{n}{k}} \sum_{(n,k)} \frac{W_i - \rho_i}{p} h^{(\omega_i)}(Z_{i_1}, ..., Z_{i_k})}{\sqrt{k^2 \zeta_{1,\omega}/n + \zeta_k/N}}, \quad D = \frac{1}{\binom{n}{k}} \sum_{(n,k)} \frac{W_i - \rho_i}{p} h^{(\omega_i)}(Z_{i_1}, ..., Z_{i_k})$$

Again, we only need to bound $P\left(z \le \frac{A}{\sqrt{\frac{k^2 \zeta_{1,\omega}}{n} + \frac{\zeta_k}{N}}} \le z + |C|\right)$. Let $0 < \eta_0 < \frac{1}{2}$, by Jensen's

inequality, we have

$$\begin{aligned} P(|C| \ge N^{\eta_0 - 1/2}) &\le N^{\frac{1}{2} - \eta_0} \mathbb{E}[|C|] \\ &\le N^{\frac{1}{2} - \eta_0} \sqrt{\mathbb{E}[|C|^2]} \end{aligned} \tag{18}$$

Note that Ghosal & Hooker (2021) illustrates that these two selection schemes (sampling without replacement and Bernoulli sampling) are asymptotically the same. More specifically, one can show that

$$\mathbb{E}[|D|^2] = \mathbb{E}\left[\left(\frac{1}{N} \sum_i (W_i - \rho_i) h^{(\omega_i)}(\mathbf{Z}_i)\right)^2\right] = K\left[\frac{1}{N} - \frac{1}{\binom{n}{k}}\right]$$

where $\mathbf{Z}_i$ represents the $i$-th subsample from $\mathbf{Z} = (Z_1, ..., Z_n)$ and $K = \mathbb{E}[(h^{(\omega_i)}(\mathbf{Z}_i))^2] = Var(h^{(\omega_i)}(\mathbf{Z}_i)) = \zeta_k$ since we assume that $\theta = 0$.

It follows that

$$\begin{aligned} P(|C| \ge N^{\eta_0 - 1/2}) &\le N^{\frac{1}{2} - \eta_0} \left(K(\frac{1}{N} - \frac{1}{\binom{n}{k}})\right)^{\frac{1}{2}} \left(\frac{k^2 \zeta_{1,\omega}}{n} + \frac{\zeta_k}{N}\right)^{-\frac{1}{2}} \\ &= (K(1-p))^{\frac{1}{2}} \left(\frac{N^{2\eta_0} k^2 \zeta_{1,\omega}}{n} + \frac{N^{2\eta_0} \zeta_k}{N}\right)^{-\frac{1}{2}} \\ &= (K(1-p))^{\frac{1}{2}} \left(\frac{k^2 \zeta_{1,\omega}}{n/N^{2\eta_0}} + N^{2\eta_0 - 1} \zeta_k\right)^{-\frac{1}{2}} \\ &\le (K(1-p))^{\frac{1}{2}} \left(\frac{k^2 \zeta_{1,\omega}}{n/N^{2\eta_0}}\right)^{-\frac{1}{2}} \\ &= (K(1-p))^{\frac{1}{2}} \left(\frac{n/N^{2\eta_0}}{k^2 \zeta_{1,\omega}}\right)^{\frac{1}{2}} \\ &= \left(\frac{n}{N^{2\eta_0}} \frac{(1-p)\zeta_k}{k^2 \zeta_{1,\omega}}\right)^{\frac{1}{2}} \end{aligned} \tag{19}$$

Eventually, we can bound $P\left(z \le \frac{A}{\sqrt{\frac{k^2 \zeta_{1,\omega}}{n} + \frac{\zeta_k}{N}}} \le z + |C|\right)$ as follows:

$$\begin{aligned} P\left(z \le \frac{A}{\sqrt{\frac{k^2 \zeta_{1,\omega}}{n} + \frac{\zeta_k}{N}}} \le z + |C|\right) &\le P\left(z \le \frac{A}{\sqrt{\frac{k^2 \zeta_{1,\omega}}{n} + \frac{\zeta_k}{N}}} \le z + |C|, |C| \le N^{-1/2}\right) + P(|C| \ge N^{-1/2}) \\ &\le P\left(z \le \frac{A}{\sqrt{\frac{k^2 \zeta_{1,\omega}}{n} + \frac{\zeta_k}{N}}} \le z + N^{-1/2}\right) + \left(\frac{n}{N^{2\eta_0}} \frac{(1-p)\zeta_k}{k^2 \zeta_{1,\omega}}\right)^{\frac{1}{2}} \\ &\le \varepsilon_3 + P(z \le Z \le z + N^{-1/2}) + \left(\frac{n}{N^{2\eta_0}} \frac{(1-p)\zeta_k}{k^2 \zeta_{1,\omega}}\right)^{\frac{1}{2}} \\ &\le \varepsilon_3 + \frac{1}{\sqrt{2\pi}} N^{-1/2} + \left(\frac{n}{N^{2\eta_0}} \frac{(1-p)\zeta_k}{k^2 \zeta_{1,\omega}}\right)^{\frac{1}{2}} \end{aligned} \tag{20}$$

Note that $\varepsilon_3$ also includes terms of order $N^{-1/2}$ and therefore

$$
\sup_{z \in R} \left| P\left( \frac{U_{n,k,N,\omega}}{\sqrt{k^2 \zeta_{1,\omega}/n + \zeta_k/N}} \right) - \Phi(z) \right| \leq \sup_{z \in R} \left| P(\frac{A}{\sqrt{\frac{k^2 \zeta_{1,\omega}}{n} + \frac{\zeta_k}{N}}} \leq z) - \Phi(z) \right|
$$

$$
+ \sup_{z \in R} \left| P(U_{n,k,N,\omega} \leq z) - P(\frac{A}{\sqrt{\frac{k^2 \zeta_{1,\omega}}{n} + \frac{\zeta_k}{N}}} \leq z) \right|
$$

$$
\leq \varepsilon_3 + \varepsilon_3 + \frac{1}{\sqrt{2\pi}} N^{-1/2} + \left[ \frac{n}{N^{2\eta_0}} \frac{(1-p)\zeta_k}{k^2 \zeta_{1,\omega}} \right]^{\frac{1}{2}}
$$

$$
= \tilde{C} \left\{ \frac{\mathbb{E}|g|^3}{n^{1/2} \zeta_{1,\omega}^{3/2}} + \frac{\mathbb{E}|h|^3}{N^{1/2} (\mathbb{E}|h|^2)^{3/2}} \right.
$$

$$
+ \left[ \frac{k}{n} \left( \frac{\zeta_k}{k\zeta_{1,\omega}} - 1 \right) \right]^{1/2} + \left( \frac{k}{n} \right)^{1/3} + N^{-\frac{1}{2}+\eta_0}
$$

$$
\left. + \left[ \frac{n}{N^{2\eta_0}} \frac{(1-p)\zeta_k}{k^2 \zeta_{1,\omega}} \right]^{\frac{1}{2}} \right\}
$$

(21)

where $\tilde{C}$ is a positive constant.

## A.9 TUNING RESULTS FOR TWO EXAMPLES

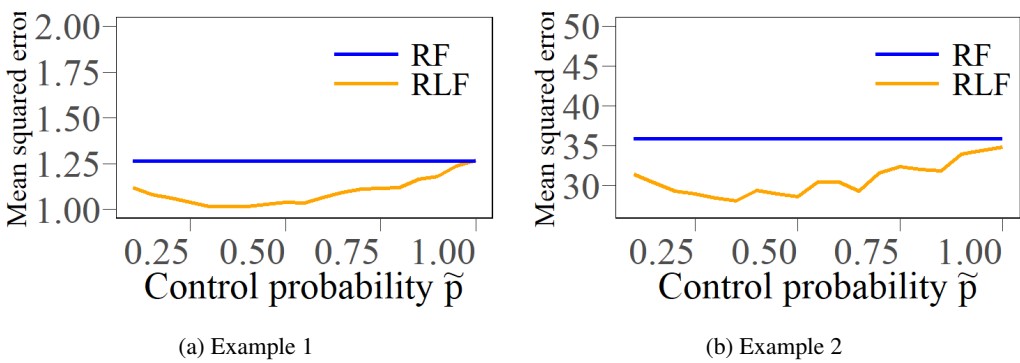

(a) Example 1           (b) Example 2

Figure S4: Test MSE curve as function of control probability $\tilde{p}$ in two examples

Table S5 : Top 10 tuned models for RLF and RF for Example 1

| Rank of Validation MSE | RF | $\alpha$ | $M_{node}$ | $M$ | RLF | $\tilde{p}$ | $M_{local}$ |
|---|---|---|---|---|---|---|---|
| 1 | 1.308092 | 0.5 | 15 | 100 | 1.081210 | 0.4 | 10 |
| 2 | 1.309780 | 0.5 | 5 | 50 | 1.085926 | 0.4 | 20 |
| 3 | 1.313989 | 0.5 | 15 | 200 | 1.088091 | 0.4 | 50 |
| 4 | 1.315487 | 0.63 | 5 | 200 | 1.117446 | 0.6 | 50 |
| 5 | 1.316665 | 0.5 | 5 | 100 | 1.125599 | 0.6 | 10 |
| 6 | 1.316748 | 0.63 | 10 | 50 | 1.127653 | 0.6 | 20 |
| 7 | 1.320919 | 0.8 | 10 | 200 | 1.128331 | 0.2 | 10 |
| 8 | 1.326341 | 0.5 | 10 | 50 | 1.150248 | 0.2 | 20 |
| 9 | 1.327373 | 0.8 | 5 | 150 | 1.164717 | 0.2 | 50 |
| 10 | 1.327539 | 0.63 | 10 | 150 | 1.197099 | 0.8 | 20 |

Table S6 : Testing MSE under optimal model for Example 1:

| | Bset RF | Best RLF |
|---|---|---|
| Testing MSE | 1.283 | 1.018 |

Table S7 : Top 10 tuned models for RLF and RF for Example 2

| Rank of Validation MSE | RF | $\alpha$ | $M_{node}$ | $M$ | RLF | $\tilde{p}$ | $M_{local}$ |
|---|---|---|---|---|---|---|---|
| 1 | 33.18551 | 0.63 | 15 | 100 | 28.48683 | 0.4 | 10 |
| 2 | 34.40944 | 0.5 | 15 | 50 | 29.34517 | 0.2 | 10 |
| 3 | 34.53584 | 0.63 | 5 | 200 | 29.53972 | 0.4 | 20 |
| 4 | 34.61853 | 0.63 | 10 | 100 | 29.72733 | 0.4 | 50 |
| 5 | 34.64231 | 0.63 | 5 | 50 | 29.98644 | 0.2 | 20 |
| 6 | 34.76124 | 0.8 | 5 | 50 | 30.22297 | 0.6 | 10 |
| 7 | 34.78667 | 0.63 | 10 | 150 | 30.31668 | 0.2 | 50 |
| 8 | 34.84602 | 0.5 | 15 | 150 | 30.33824 | 0.6 | 50 |
| 9 | 34.99652 | 0.8 | 5 | 150 | 30.94450 | 0.6 | 20 |
| 10 | 35.19693 | 0.63 | 15 | 150 | 32.03192 | 0.8 | 10 |

Table S8 : Testing MSE under optimal model for Example 2:

| | Best RF | Best RLF |
|---|---|---|
| Testing MSE | 35.17 | 29.87 |

**Tuning procedure:** For each example, we generated 3000 samples from the corresponding model and divide in into three parts for training, validation and testing. The ratio is 6:2:2. For RF, we set subagging ratio $\alpha \in \{0.5, 0.63, 0.8\}$, minimal node size $M_{node} \in \{5, 10, 15\}$ and number of trees $M \in \{50, 100, 150, 200\}$. For RLF, we keep $M = 100$ and $\alpha = 0.63$ all the time for efficiency. We set $\tilde{p} \in \{0.2, 0.4, 0.6, 0.8\}$ and $M_{local} \in \{10, 20, 50\}$ which are two new parameters introduced in RLF. Fig. S4(a) and S4(b) are Test MSEs for RLF as functions of $\tilde{p}$ with $M = 100, M_{local} = 10$. For RF, we use the default setting.

### A.10 Extra experiments with tuned RLF and RF

We performed extra experiments on 26 datasets (due to the time efficiency of parameter tuning) to compare best RF and best RLF.

We performed 5-fold cross validations to ensure 20% of observations are used as testing set. For the rest of 80% points, we further randomly pick 25% of them as validation set, which is used to select best models among parameter space. As a result, the ratio of training, validation and testing is 6:2:2.

Tuning parameters for RLF: $\tilde{p} \in \{0.4, 0.6, 0.8\}$. We set $M = 100, M_{local} = 10, M_{node} = 5$ all the time for RLF due to the time efficiency. Tuning parameters for RF: $M \in \{50, 150, 200\}$, $M_{node} \in \{5, 10, 15\}$. Other parameters for RLF and RF follow the default value. Note that the implementation of RF in R doesn't have parameter of tree depth but we can control and depth of tree by the value of minimal size of node $M_{node}$.

