# OpenReview forum: "Riemann-Lebesgue Forest for Regression"
_ICLR.cc/2025/Conference — ICLR 2025 Conference Withdrawn Submission_

### Official Review · Reviewer_Fs4M · 2024-10-29

**Soundness:** 3
**Presentation:** 2
**Contribution:** 1
**Rating:** 5
**Confidence:** 3

**Summary:**

The author(s) present a new ensemble learning method named Riemann-Lebesgue Forest (RLF) for regression tasks. Traditional random forest methods use "Riemann" sense partitioning to approximate a function that best fits the observed data. The proposed method applies a "Lebesgue" sense partitioning based on the range of the response variable Y, aiming to achieve a more accurate model with reduced error. Additionally, the paper addresses the proposed method from a theoretical points, including proofs of asymptotic normality and a complexity analysis.

**Strengths:**

- RLF introduces a new way of dividing data in regression trees, called "Lebesgue" cutting, which enhances accuracy in situations where traditional random forests (RF) struggle, especially in high-noise or low-data scenarios.
- The paper explains the main theories behind RLF, giving a solid mathematical base.

**Weaknesses:**

- The paper lacks a publicly accessible GitHub repository or similar platform for the proposed methods.
- RLF generally requires more wall-clock time than RF, especially for large datasets, and does not show significant performance improvements in terms of time complexity.

**Questions:**

RLF is more accurate but takes more time to run. Is this extra accuracy worth it for real-life use, especially when time is important? It would help to explain and show the level of contribution the approach makes to accuracy.

---

> ### Author Response · Authors · 2024-11-13
>
> Dear reviewer, we thank you for your thoughtful feedback. We are encouraged that you agree that RLF enhances accuracy in situations where traditional random forests (RF) struggle. We will address your main concerns as follows.
>
> **W1: The paper lacks a publicly accessible GitHub repository or similar platform for the proposed methods.**
>
> **Response:**
>
> In lines 97-98, we mentioned that the raw code of implementation of RLF is enclosed in the supplementary material. You can download the supplementary material and install the R package of RLF locally in the R studio (See readme file in the supplementary material).  The R codes for simulations and experiments  in manuscript can also be found in supplementary material.
>
> **W2: RLF generally requires more wall-clock time than RF, especially for large datasets, and does not show significant performance improvements in terms of time complexity.**
>
> **Response**
>
> Note that Table 1 in the original manuscript shows that RLF with data-driven $\tilde{p}$ outperforms RF in 20 datasets where eight of them are significant.  Table 2 shows that RLF with tuned $\tilde{p}$ outperform RF in 23 datasets where twelve of them are statistically significant.  Those results indeed demonstrate the significant superiority of RLF over RF, with the price of time efficiency.  However, RF only statistically outperform RF in two datasets.
>
> On the other hand, the  time complexity of RLF is mainly due to the local random forest and the nature of tree ensemble method. To reduce the time complexity of RLF, we can construct single RLT in parallel and employ a more efficient local model for nodes with Lebesgue type splitting, which will be our future work.
>
> **Question: "...Is this extra accuracy worth it for real-life use, especially when time is important?...**
>
> **Response**
>
> As we explained in response to W2, the time complexity can be reduced from parallelization and more efficient local models.
> As to the reduction percentage of MSE, Table 1 shows that RLF reduces testing MSE by 15\% on average, which is a decent improvement. Similar enhancement can be seen in Table 2 (with tuned $\tilde{p}$).
>
> In lines 493-497, we discussed that  time efficiency is the main limitation of RLF. For prediction part, we
> employ random forest locally which is powerful but time consuming for large dataset. The time efficiency of
> current RLF can be improved by employing less number of subtrees in local forest and setting larger
> value of control probability $\tilde{p}$ so that each RLT won’t perform too many Lebesgue splittings.
>
> Additionally, for the current version of RLF, we still see a significant improvement of accuracy in datasets with thousands instances and the time efficiency of RLF is acceptable (Table 1 and Table S4). For some medical or chemistry experiments , we typically have small number of observations due to the constraints from  patients or investment. In those cases we believe the current RLF will have some value in providing the lower prediction error under acceptable time constraint.
>
> This paper focuses primarily on the offline performance of RLF, i.e we have sufficient time budget. The online application of RLF where the time constraints can be tight is another interesting direction. The more efficient implementation of RLF such as parallelization is an engineer problem, which is our future work and out of the scope of current paper.

---

### Official Review · Reviewer_78aY · 2024-11-03

**Soundness:** 3
**Presentation:** 3
**Contribution:** 3
**Rating:** 8
**Confidence:** 3

**Summary:**

The authors introduce Reimann-Lebesgue Forest (RLF), a new ensemble method for regression that mimics Lebesgue integration by occasionally splitting nodes based on the response variable Y. They also establish the asymptotic normality of RLF and demonstrate its competitive performance against Random Forests on both simulated and real-world datasets.

**Strengths:**

1. The authors propose an interesting “Riemann-Lebesgue” splitting criterion for CART that directly incorporates the response variable splitting, making it a novel approach worth exploring.
2. The authors provide a solid theoretical justification for their approach.
3. The approach demonstrates effectiveness compared to traditional Random Forests across various synthetic and real datasets.

**Weaknesses:**

1. The notation in Section 2.1 could be clarified. In lines 112 and 133, the dimension is denoted as d, whereas in lines 127 and 136, it appears to be represented by p. For better readability, it would be preferable to use consistent notation for dimension throughout. Additionally, since  m_{try}  is an integer, the set  C_R  in line 135 is not well defined.
2. I don’t believe it’s impractical to keep partitioning in practice, as suggested in line 53. For example, the default min_samples_leaf parameter in Scikit-Learn’s Random Forest Regressor is set to 1 (M_node = 1), which allows the trees to grow very deep. A follow-up concern is whether the difference between RLF and RF would diminish if RF trees were allowed to go deeper, while it runs faster than RLF. Specifically, the paper used a minimum node size of M_node = 5 for real datasets, as shown in Appendix A.10. What would happen if M_node were reduced to 1?
3. While Section 3.3 addresses the training complexity, I believe the contrast in prediction complexity is even more significant. Has any analysis been conducted on the prediction time complexity?
4. The introduction of ‘Lebesgue’ partitioning could complicate interpretation, particularly for calculating values like TreeSHAP [1], although the paper’s primary focus is on predictive performance.

Some typos:

1. In algorithm 1, “repeat steps 1 - 15” in line 8 and line 12. The final line also has an alignment issue.
2. Line 658 “glocal” trees
3. Table S4 shows the SE of running time for RF on dataset FF as 5432.01 which is quite high, I guess the author used the MSE in Table 1.

**Questions:**

When noise levels are high, splitting on signal features tends to be less effective, and intuitively, ‘Riemann’ type splitting might also be expected to underperform. Could the authors provide intuition for why ‘Riemann’ type cuttings achieve better performance at higher noise levels, as suggested in lines 434-435?

---

> ### Author Response · Authors · 2024-11-13
>
> Dear reviewer, we appreciate your careful reading and detailed comments on our work. We are glad that you found that  “Riemann-Lebesgue” splitting is interesting and worth exploring. We will address your concerns as follows.
>
> **W1: The notation in Section 2.1 could be clarified.**
>
> **Response**
>
> We have unified the notation for the feature space dimension in modified manuscript. We denote $d$ as the dimension of feature space. We also defined $C_R$ as $\\{ (j,z^{(j)}): j\in \\{p_1,...,p_{m_{try}}\\},z^{(j)}\in \mathbf{X}^{(j)} \\}$ where the set $\{p_1,...,p_{m_{try}} \}$ represents randomly picked feature index in CART. See line 136-137 in modified manuscript.
>
> **W2: "... What would happen if M\_node were reduced to 1?""**
>
> **Response**
>
> For time complexity, we didn't set $M_{node}$ too small as too small $M_{node}$ may slow down the training and inference speed of RLF and RF. On the other hand, setting $M_{node}$ to 1 is more likely to make fitted RF and RLF overfitting. In practice, $M_{node}=5$ seems to be enough in most cases and we anticipate that setting $M_{node}=1$ won't make big difference in most cases.
>
> We  performed extra experiments (10-folds cross validation) on few datasets to see the effect of $M_{node}=1$. (We ignored few time consuming datasets). See following table. Other parameter settings follow the Appendix A.6. The following table shows that RLF still reduces
> testing MSE by 15% on average which is similar to the improvement in Table 1.
>
> |Dataset| RLF(100,10,0.8)| RF(100)|
> |-|-|-|
> |FF|**4271.02**$\pm5575.50$| 4333.86$\pm 5418.68$|
> |SP|**1.74**$\pm 0.80$|1.81$\pm 0.80$|
> |EE|**0.66**$\pm 0.27$| 1.23$\pm  0.31$|
> |CAR|**4851917**$\pm 807664.2$|5311459$\pm 832711.8$|
> |QSAR|**0.75**$\pm 0.13$|0.85$\pm0.13$|
> |CCS|22.39$\pm 4.91$|26.12$\pm 4.63$|
> |SOC|**432.66**$\pm 181.50$|565.87$\pm 197.31$|
> |GOM|**240.57**$\pm  30.33$|245.23$\pm 28.96$|
> |SF|0.74$\pm 0.13$|0.76$\pm 0.12$|
> |ASN|**6.24**$\pm 0.94$|12.71$\pm  1.28$|
> |WINER|0.32$\pm 0.033$|0.32$\pm 0.033$|
> |AUV|4253840$\pm 753585.4$|8929430$\pm  1136452$|
> |SG|0.013$\pm 0.0045$|0.014$\pm  0.0047$|
> |ABA|4.63$\pm 0.58$|4.61$\pm 0.56$|
> |WINEW|0.35$\pm 0.022$|0.36$\pm0.0019$|
> |CPU|5.49$\pm 0.49$|5.93$\pm 0.59$|
> |KRA|0.017$\pm 0.0007$|0.021$\pm 0.0008$|
> |PUMA|**0.00047**$\pm 1.6E-05$|0.00051$\pm 1.37E-05$|
> |GS|0.00011$\pm 3.75E-06$|0.00015$\pm 4.3E-06$|
> |BH|0.0099$\pm 0.01$|0.011$\pm 0.012$|
> |NPP|**4.43E-07**$\pm 5.32E-08$|5.81E-07$\pm 7.18E-08$|
> |MH|**0.020**$\pm 0.0009$|0.022$\pm 0.0011$|
>
> Bolded text means statistically significant.
>
> **W3:"... Has any analysis been conducted on the prediction time complexity?"**
>
> **Response**
>
> We indeed found that the prediction time of RLF is more important in practice. The rigorous analysis is too complicated as RLF involves the randamness of using "Riemman" or "Lebesgue" type splitting.  However, we can do some preliminary analysi here.
>
> Suppose at certain node A (size m) in RLF, the time complexity of determining which direction to go is just O(1) when this node employed "Riemann" type splitting as we only need to compare selected feature with threshold value once.  If this node used "Lebesgue" type splitting, we may need to compare selected feature with threshold value d(m)*M\_local times as the new incoming point need to go through the local randomforest, where $d(m)$ is the depth of each tree in local forest at node A, which increases the inference time of RLF.
>
> This observation would explain the relatively low time efficiency of RLF in Table S4.
>
> **W4:"The introduction of ‘Lebesgue’ partitioning could complicate interpretation, particularly for calculating values like TreeSHAP [1], although the paper’s primary focus is on predictive performance."**
>
> **Response**
>
> We agree that ‘Lebesgue’ partitioning could make  interpretation complicated. How to balance the predictive performance and interpretation will be our future work. In fact, we can control the number of "Lebesgue" type splitting with the $\tilde{p}$, which is the novel randomness introduced in RLF.
>
> Additionally, the tuning process of RLF in example 1 and example 2 provides us an insight of the structure of response. If tuned RLF tends to use more "Riemann" type splitting, we can expect high noise level in response. If  tuned RLF tends to use more "Lebesgue" type splitting, it's possible that we have complicated structure in response (e.g mixture distribution).  Those information could be helpful in the intepretation of the dataset and model.
>
> **Some typos**
>
> We have corrected the typos listed in weakness in modified manuscript. The corrected texts are highlighted in blue. We thank again for your proof reading!

---

> > ### Author Response · Authors · 2024-11-13
> >
> > **Question: "When noise levels are high, splitting on signal features tends to be less effective, and intuitively, ‘Riemann’ type splitting might also be expected to underperform. Could the authors provide intuition for why ‘Riemann’ type cuttings achieve better performance at higher noise levels, as suggested in lines 434-435?"**
> >
> > **Response**
> >
> > This is a very interesting question.  One possible reason comes from the overfitting due to ‘Lebesgue’ type splittings. When noise level in response is high, cutting point in response can be misled by noises in $Y$.
> >
> > It is also true that ‘Riemann’ type splitting might be expected to underperform in high noise level cases and splitting on signal features can be  ineffective as well. However, splitting on $Y$ can be even worse than  splitting on signal features when noise level is high as ‘Lebesgue’ type splittings are \textbf{directly} affected by noises in response.
> >
> > As a result, the tuned RLF tends to less number of ‘Lebesgue’ type splittings to avoid overfitting. In other words, tuned RLF tends to use more ‘Riemann’ type splittings when the noise level is high.

---

> > > ### Comment · Reviewer_78aY · 2024-11-25
> > >
> > > I thank the authors for the additional experiments and detailed responses. While the local forest approach is somewhat controversial, I agree that it’s not a major issue since we only need to perform better than a coin flip (although the timing is still a concern). I am raising my score to 8 for the following reasons:
> > >
> > > 1. The idea is novel to me, and while I don’t expect it to be SOTA immediately, it has the potential to inspire other research in the field.
> > > 2. The method shows promise in capturing unobserved confounders, I suddenly realized the compromised interpretation can add to this aspect. Expanding the experiments to datasets like Example 2 would be particularly interesting, especially to the causal inference community.
> > >
> > > BTW, another typo: Line 187 it’s - its

---

> > > > ### Author Response · Authors · 2024-11-25
> > > >
> > > > Dear reviewer,
> > > >
> > > > Thank you for raising your score and we are glad that our response addressed your main concerns. We have corrected the typo in line 187. See the latest version of pdf file.
> > > >
> > > > It's also interesting that you pointed out the potential application of RLF to the causal inference community. We are excited to pursue this direction in our future work.

---

### Official Review · Reviewer_aeZJ · 2024-11-04

**Soundness:** 2
**Presentation:** 2
**Contribution:** 2
**Rating:** 3
**Confidence:** 4

**Summary:**

The paper considers learning a variation of Random Forests, where each decision tree is learned using CART-type tree induction, but instead of considering only input features to split, it considers the target output also as a splitting criteria. But during the test time, we need the value of the target during inference (at decision nodes that splits on the target), and those are computed (predicted) using Random Forest (which will also be trained locally at decision nodes that splits on target). The paper has some theoretical analysis, and experimental comparison with Random Forests.

**Strengths:**

Using target values as one of the splitting criteria in tree learning is an interesting idea.

**Weaknesses:**

1. The novelty of the paper is not on learning a forest in a novel way but rather learning a single decision tree using splits on the target. As such, the new tree learning method should be compared on its own with other tree learning methods such as CART, C5.0, GUIDE, TAO and PyDL8.5. According to theorem 3.1, shouldn't it be better than CART?
2. Using local Random Forest to predict the output that is used at a decision node splitting on the target is quite a heuristic approach. Is there any theory or intuitive motivation behind it?
3. No comparison with gradient boosting. XGBoost and LightGBM are the widely established methods in tree ensembles. Given the fact that tree ensembles are mostly evaluated empirically, having a comparison with gradient boosting is important.
4. Theoretical analysis of section 3 looks very complicated, especially the bounds of theorem 3.2. They do not seem to add any practical value.

**Questions:**

No questions.

---

> ### Author Response · Authors · 2024-11-13
>
> Dear reviewer, we are encouraged you found that using target values as one of the splitting criteria in tree learning is an interesting idea.  We will  address your concerns as follows.
>
> **W1: The novelty of the paper is not on learning a forest in a novel way but rather learning a single decision tree using splits on the target.**
>
> **Response:**
>
> As we stated in abstract and introduction, the main goal of this paper is to introduce a new type of forest method with the novel base tree learner. We primarily focused on the predictive performance of RLF rather than a single RLT.
>
> Theorem 3.1 only implies the better training error of RLT not the testing error. In fact, the use of Lebesgue type splitting leads to the overfitting. Comparing the performance with other single tree learning is less useful w.r.t to our ultimate goal.
>
> To control the potential overfitting resulted from Lebesgue type splitting, we made the following efforts:
>
> 1) We introduced the control probability $\tilde{p}$ to control the number of Lebesgue type splittings in each Riemann-Lebesgue tree. (Line 118-120). We also provided two types of control probability $\tilde{p}$.
>
> One is a data-driven $\tilde{p}$, which is defined in line 118. The inequality $L(j^*,z^*) \leq \tilde{L}(z_{L}^{*})$ tells us that the value of $\tilde{p}$ is at least $1/2$, indicating that there won't be too many Lebesgue type splittings in each single tree. As a result, the effect of potential overfitting is limited.
>
> The other is tuned $\tilde{p}$, which is briefly introduced in section 4.3. That is to say, we set $\tilde{p}$ be a fixed value for all nodes in a RLT. The optimal value of $\tilde{p}$ can be given by the tuning procedure (cross-validation).  Fig 3(a), S4(a) and S4(b) exhibit the potential benefit of tuning control probability $\tilde{p}$.   Extra experiments of RLF with tuned $\tilde{p}$ (see Table 2) further demonstrate the superiority of RLF with tuned $\tilde{p}$.
>
>
> 2) The ensemble step in Riemann-Lebesgue Forest is also critical in reducing the overall overfitting as the ensemble itself is an efficient way to reduce the variance of base learner. Actually, the traditional random forest is a good example of using ensemble method to reduce the overfitting resulted from unpruned decision tree.
>
> In summary, the flexibility of control probability $\tilde{p}$ and the ensemble step in RLF both contribute to the improvement of overfitting resulted from Lebesgue type splitting.
>
> **W2: Intuition of using random forests as the local model**
>
> **Response:**
>
> In lines 189-192, we mentioned that Linear regression is one of the candidates for the local model. However, it is unstable when the sample size is relatively small. Another choice is the K-Nearest Neighborhood algorithm (KNN) whose performance relies on the distance function we used which can be unstable in high-dimensional cases.
>
> In this paper, we choose random forest as the local model to obtain an estimate of the response value
> of a new incoming point since random forest is parameter free and robust for small sample size.
>
> **W3: No comparison with gradient boosting**
>
> **Response:**
>
> We performed extra experiments (10-folds crossvalidation) comparing the **testing MSE** of RLF,GradientBoosting(GB) and XGboost. The following table illustrates that RLF is comparable XGboost in general.  Number of base tress in GB and XGboost are 1000 and subagging ratio is 0.8. Other parameters are by default. Number of RLTs is 100 in RLF and local RF in RLF has 10 subtrees and $\tilde{p}=0.8$.
>
> |Dataset| RLF(100,10,0.8)| GB(1000,0.8)|XGboost(1000,0.8)|
> |-|-|-|-|
> |FF|**4293.74**$\pm 5527.51$|4694.78$\pm 5363.52$|6624.42$\pm 5518.72$|
> |SP|**1.76**$\pm 0.79$|2.13$\pm 0.90$|2.27$\pm 0.90$|
> |EE|0.74$\pm 0.27$| 1.09$\pm  0.22$|**0.09**$\pm 0.03$|
> |CAR|**4916065**$\pm 776998.4$|8781409$\pm 1181857$|5802233$\pm 988893.7$|
> |QSAR|**0.75**$\pm 0.13$|0.85$\pm0.13$|0.97$\pm 0.17$|
> |CCS|23.72$\pm 4.38$|28.87$\pm 3.68$|**20.05**$\pm 8.49$|
> |SOC|422.94$\pm164.72$|460.71$\pm 135.03$|**325.54**$\pm 167.66$|
> |GOM|**240.13**$\pm 28.46$|266.72$\pm 37.29$|269.68$\pm24.96$|
> |SF|0.71$\pm 0.20$|**0.61**$\pm 0.19$|0.95$\pm0.18$|
> |ASN|6.95$\pm 0.79$|20.94$\pm  2.25$|**1.58**$\pm 0.39$|
> |WINER|**0.32**$\pm 0.03$|0.41$\pm 0.038$|0.38$\pm 0.04$|
> |AUV|4324542$\pm 711281.7$|27726483$\pm  1730233$|**155060.7**$\pm 38776.38$|
> |SG|0.013$\pm 0.005$|0.016$\pm  0.005$|**0.012**$\pm0.003$|
> |ABA|**4.60**$\pm 0.58$|4.79$\pm 0.43$|5.76$\pm 0.65$|
> |WINEW|**0.36**$\pm 0.02$|0.50$\pm0.03$|0.40$\pm 0.02$|
> |CPU|**5.45**$\pm 0.49$|6.25$\pm 0.47$|6.00$\pm 1.45$|
> |KRA|0.017$\pm 0.0008$|0.040$\pm 0.0008$|**0.016**$\pm0.0007$|
> |PUMA|**0.00047**$\pm 0.0008$|0.00112$\pm 0.0008$|0.00056$\pm0.0007$|
> |GS|0.00011$\pm 3.81E-06$|0.0003$\pm 9.69E-06$|**8.97E-05**$\pm2.42E-06$|
> |BH|0.01$\pm 0.01$|0.014$\pm 0.008$|**0.004**$\pm0.006$|
> |NPP|**4.81E-07**$\pm 5.12E-08$|3.90E-05$\pm 1.44E-06$| 8.76E-07$\pm8.11E-08$|
> |MH|**0.020**$\pm 0.0009$|0.032$\pm 0.0017$|0.023$\pm0.001$|

---

> ### Author Response · Authors · 2024-11-13
>
> **W4: Practical value of Theoretical analysis of section 3**
>
> **Response:**
>
> Theorem 3.1 provides a theoretical explanation of why RLF is expected to have smaller mean squared error
> than RF after the ensembling. The critical point is the introduction of Lebesgue type splittings in building a single CART tree.
>
> The asymptotic normality induced by  Theorem 3.2 provides a useful tool for statistical inference of RLF. For example, [1] first constructs confidence intervals and hypothesis test w.r.t original random forest.
>
> The Berry-Esseen bounds of RLF generalizes all possible convergence rates under different parameter settings which gives a guidance in choosing appropriate formula in practice.  The asymptotic result in small-N setting  provide  a theoretical support for people employing less number of base learners as long as the subsample size k is appropriately designed.
>
> The time complexity analysis in section 3.4 provides readers the factors effecting the time efficiency of RLF, which are helpful for future optimized implementation.
>
> [1] Lucas Mentch and Giles Hooker. Quantifying uncertainty in random forests via confidence intervals
> and hypothesis tests. J. Mach. Learn. Res., 17(1):841–881, jan 2016. ISSN 1532-4435.

---

> > ### Comment · Reviewer_aeZJ · 2024-11-25
> >
> > I appreciate the authors' detailed response. However, I remain unconvinced about the general novelty of the paper in the context of learning tree ensembles, the heuristic approach of using random forests as local models, and the practical significance of the provided theorems. As such, I maintain my original score.

---

> > > ### Author Response · Authors · 2024-11-25
> > >
> > > Dear reviewer,
> > >
> > > Thank you for the response. We respect your comments. However, we'd like to emphasize few points.
> > >
> > >
> > > ## 1.Novelty of RLF
> > >
> > > The novelty of learning partitions from response variable in RLF lies in its analogue of Lebesgue partitions of a measureable function (a standard fact in a real analysis course).  In some real cases, there might be some noisy features in X space. Simply split the feature space from X may make fitted model (e.g. traditional RF) biased.  The numerical example of sparse model in section  4.1 demonstrates the superiority of RLF when we have too many redundant X variables.  **Splitting from response variable** enables us to  exploit information hidden in the response rather than predictors only, which can be helpful in sparse models. Note that many tree ensemble methods only focus on partitioning X space.
> > >
> > > Additionally,  as we stated in the first two points of the meta rebuttal.  The introduction of the probability, which controls the type of splitting (Riemann or Lebesgue) at each non-terminal node, makes the RLF more flexible in detecting complex patterns in response Y.  Similar to the  randomness resulted from "feature bagging" in the traditional RF, our RLF introduces a type of "split type randomness".
> > >
> > > In section 4.3, we provided two one-dimensional exmaples illustrating the flexibility of RLF with tuned $\tilde{p}$
> > > . More specifically, in datasets with small signal-to-noise ratio (example1), RLF favors more Riemann type splitting. This is intuitive as we typically don't want to split from Y if it's too noisy. On the other hand, when the response Y has mixture distribution (example 2), experiments show that tuned RLF favors more Lebesgue type splitting. This also makes sense as when the response has complicated pattern, the Riemann type cutting might be too simple to capture the pattern of response Y.  Those examples demonstrate the cases when the RLF is expected to perform better than traditional RF.
> > >
> > > ## 2. The practical significance of the provided theorems, especially Thm 3.2
> > >
> > > In probability theory, the inequality derived in Thm 3.2 is called the Berry-Esseen bound of RLF. The establishment of the Berry-Esseen bound proves the asymptotic normality of RLF when the data size is large.  Once we obtain the normality of a model, many statistical analysis tools such as confidence interval and hypothesis test are available.
> > >
> > > Similar results for tranditional RF have been establised by [1], who derived Berry-Esseen bounds of traditional RF and invented confidence intervals to  allow users to investigate prediction variability at particular points or regions. This can be viewed as a way to measure of how much the accuracy of predictions at that point is due to chance.
> > >
> > > They also developed a hypothesis testing w.r.t feature significance in [1], which  can be extremly useful in sparse model regression problem as we can rule out many redundant features based on the results of hypothesis testing.
> > >
> > > We believe readers who are interested in developing statistcal inference / interpretation tools for tree ensembling methods can benefit from our Thm 3.2
> > >
> > > [1] Lucas Mentch and Giles Hooker. Quantifying uncertainty in random forests via confidence intervals and hypothesis tests. J. Mach. Learn. Res., 17(1):841–881, jan 2016. ISSN 1532-4435
> > >
> > > ## 3. Intuition behind local random forests ##
> > >
> > > The intuition of using small random forests stems from the fact that many simple parametric methods such as regressions, KNN are unstable in small sample size. In other words, when we grow a tree deeply, the ability of those methods telling which direction a new point should go will be worse than non-parametric methods, such as random forests.
> > >
> > > On the other hand, because of the flexibility of random forests, we can not only adjust the number of subtrees in a local random forest but also set small subtree depth  to balance the time efficiency and prediction performance of RLF.
> > >
> > > Of course we believe there exists other choices of local models which might be more time efficient than local random forests. This will be our future work.

---

### Official Review · Reviewer_gNLn · 2024-11-04

**Soundness:** 3
**Presentation:** 1
**Contribution:** 2
**Rating:** 5
**Confidence:** 4

**Summary:**

The paper introduces a new ensemble regression method called the Riemann-Lebesgue Forest (RLF), which leverages a novel tree structure, the Riemann-Lebesgue Tree (RLT). RLF improves traditional Random Forests by utilizing "Lebesgue-type" partitioning, which splits based on response values rather than predictor values alone. This approach, combining both Riemann and Lebesgue partitioning, is shown to reduce variance in predictions and improve mean squared error, particularly in sparse or noisy data scenarios. The authors provide theoretical analyses, demonstrating the asymptotic normality of RLF and evaluating its performance on both simulated and real-world data, highlighting RLF's competitive edge in regression tasks over conventional method.

**Strengths:**

- **Originality**: The paper introduces a novel ensemble method, the Riemann-Lebesgue Forest (RLF), which creatively combines Riemann and Lebesgue partitioning within decision trees. This innovative approach, applying response-based splits (Lebesgue-type) rather than traditional feature-based splits, offers a fresh perspective on ensemble methods, especially in handling regression tasks in sparse models.

- **Quality**: The research is thorough, with extensive theoretical analysis and rigorous experimentation. The authors carefully examine the variance reduction and asymptotic normality properties of the RLF, providing proofs that establish the foundation of the method's performance.

- **Significance**: The introduction of the Lebesgue-type partitioning extends the applicability of ensemble methods to settings where traditional Random Forests might underperform, particularly in high-dimensional or noisy environments.

**Weaknesses:**

- **Clarity in the Splitting Criteria**: The connection between Eq. (2) (simple function approximations) and Eq. (5) (the splitting criterion) is unclear, which weakens the reader's understanding of the novel splitting mechanism. It is really needed to expand the explanation on this connection.

- **Overly Strong Claims**: The claim that "a RLT will have smaller \(L_2\) training error than an ordinary CART tree" is not straightforward and needs more explanation. Furthermore, the expected reduction in mean squared error due to bias-variance decomposition analysis does not explain the fact that the new proposed method overfits. This overfitting is usually the result of a method having high variance. This needs to be much better explained.

- **Limited Experimental Improvements**: The empirical results, while promising, show only modest improvements over traditional Random Forests.

- **Dependency on a Local Model**: The requirement of a secondary random forest model within each Riemann-Lebesgue Tree as a local model presents computational and interpretative limitations. This reliance could be explored in greater depth, addressing alternative local models or the impact of this dependency on scalability and performance. This limitation should, at least, much better discussed.

**Questions:**

Please try to answer each of the weaknesses discussed above.

---

> ### Author Response · Authors · 2024-11-13
>
> Dear reviewer, we thank you for your thoughtful feedback. We are encouraged that you found that RLF offers a fresh perspective on ensemble methods and outperforms traditional RF in high-dimensional or noisy environments. We will address your concerns as follows.
>
> **W1: Clarity in the Splitting Criteria**
>
> **Response:**
>
> The reason why we introduced simple function approximations in Eq.(2) is to give the readers who are not familiar with the idea of approximating function by Lebesgue partition a basic intuition.
>
> Eq. (5) (the splitting criterion) is a special case of the idea of Lebesgue partition, i.e we find the optimal splitting point in response which reduces the variance of the node most.  In Eq.(2) (simple function approximations) , the Lebesgue partition is naive but not useful in building a effective base learner tree in our ensemble method. That's why we further introduced Eq. (5) to embed the spirit of Lebesgue partition into building a novel base learner tree.
>
> **W2:  Overly Strong Claims:**
>
> **Response:**
>
> Theorem 3.1  showed that  the Lebesgue splitting  will lead to smaller variance of partitioned response Y in theoretical after the splitting process(training step in build a tree model). In the end, the variance, which is the L2 error at each terminal node will be lower than that of original CART. Note that in regression tree, we use variance (L2 error) as the impurity function.  This means that RLT will generate smaller training error than a ordinary CART tree given other tree parameters are the same. The training error is interpreted as bias in traditional ML theory.
>
> The general thinking is, the  set of possible partitions of current set of responses induced by the Riemann cutting of covariate $\mathbf{X}$ is a subset of that of Lebesgue cutting whose feasible set is essentially  induced by response directly. Note that  the objective functions $L(j,z)$ and $\tilde{L}(z)$ have the same form, the one with larger feasible set should have larger optimal value, i.e the larger variance (L2 error) reduction.
>
> More specifically, suppose we have $n$  points $\{(X_1,Y_1),...,(X_{i},Y_{i})\}$ whose y values are all distinct in a certain non-terminal node and both cutting types employ CART split criterion,  then Lebesgue cutting will go through all $(n+1)$ splitting points in $Y$ directly, which essentially gives the largest feasible set of optimization problem.  On the other hand, in Riemann type cutting, the space of $Y$ is indirectly partitioned by going through all possible splitting points in direction $X^{(j)}$. It's not necessary that this procedure will take care all $(n+1)$ splitting points in $Y$. For example, when $X^{(j)}$ is a  direction with only three distinct values , $z^{(j)}$ can only have four choices of splitting point which will restrict the possible values of $\bar{Y}{A_{L}}$ , $\bar{Y}{A_{R}}$  in optimizing $L(j,z)$   and leads to a smaller feasible set.
>
> The overfitting of RLT is expected if we fully or partially utilize the information from $Y$. In lines 169-171, we discussed the case of perfect fitting of Lebesgue type splitting. This phenomenon has been illustrated in Figure 3: (a) and Figure S4.
>
> To control the potential overfitting resulted from Lebesgue type splitting, we introduced the control probability $\tilde{p}$ to control the number of Lebesgue type splittings in each Riemann-Lebesgue tree (Lines 118-120) and ensemble step in reducing the overall overfitting as the ensemble itself is an efficient way to reduce the variance of base learner.
>
> The experiment results imply that  the flexibility of control probability $\tilde{p}$ and the ensemble step in RLF both contribute to the improvement of overfitting resulted from Lebesgue type splitting.
>
> **W3:  Limited Experimental Improvements**
>
> **Response:**
>
> Note that Table 1 in the original manuscript shows that RLF with data-driven $\tilde{p}$ outperforms RF in 20 datasets where eight of them are significant.  Table 2 shows that RLF with tuned $\tilde{p}$ outperform RF in 23 datasets where twelve of them are statistically significant.  Those results indeed demonstrate the significant superiority of RLF over RF, with the price of time efficiency.  However, RF only statistically outperform RF in two datasets.
>
> As to the reduction percentage of MSE, Table 1 shows that RLF reduces testing MSE by 15\% on average, which is a decent improvement. Similar performance can be seen in Table 2.

---

> > ### Author Response · Authors · 2024-11-13
> >
> > **W4: Clarity in the Splitting Criteria**
> >
> > **Response:**
> >
> > In lines 189-192, we mentioned that Linear regression is one of the candidates for the local model. However, it is unstable when the sample size is relatively small. Another choice is the K-Nearest Neighborhood algorithm (KNN).
> > However, the performance of KNN relies on the distance function we used which can be unstable in
> > high-dimensional cases.
> >
> > In this paper, we choose random forest as the local model to obtain an estimate of the response value
> > of a new incoming point since random forest is parameter free and robust for small sample size. We
> > believe there exists more efficient local models, which is our future work.
> >
> > Note that the local models in single RLT is used to tell which node to proceed to, not to tell the final prediction directly. Thus it's more important for a local model to correctly label the direction of new points rather than the simple prediction. To this sense, the local random forest  is a strong candidate compared with other traditional statistical models.

---

> > > ### Comment · Reviewer_gNLn · 2024-11-25
> > >
> > > Dear Authors,
> > >
> > > I really appreciate your thorough responses. I've been reading them as well as all the others reviews and discussions. You have clarified my main concerns. Even though,  I can't increase the score of the paper. In my completely subjective opinion, the significance and novelty of this contribution is not relevant enough to motivate its acceptance to this highly competitive venue. RFs are very well studied, the provided results here give only marginal gains in most of the cases, when compare to gradient boosting there is no really a clear advantage, your method needs a local model which can potentially introduced a lot computational complexity.

---

> > > > ### Author Response · Authors · 2024-11-25
> > > >
> > > > Dear reviewer,
> > > >
> > > > Thank you for the feedback. We are glad that our rebuttal clarified your main concerns and we also respect your opinion.
> > > >
> > > > On the other hand, we'd like to point out that the **flexibility of local random forests makes it possible to control the computation cost in some sense**. For example, we can set the number of subtrees and the depth of each subtree in a local random forest be both appropriately small to speed up the fitting and prediction procedure in RLF.  The control probability $\tilde{p}$ also plays an role in balancing the time efficiency and prediction performance of RLF. Note that we don't require the local model  to be very precise as we only need to know where shoud a point go during the prediction.
> > > >
> > > > The robustness of local random forest is our main consideration. We indeed have many fast candidates of local models such as linear regression and KNN. However, as we stated in previous rebuttal, those methods are not reliable in some cases. In practice, there is always a trade-off between time efficiency and reliability (or prediction performance).
> > > >
> > > > **As to the comparison with gradient boosting methods**, we conducted extra experiments in the response to **reviewer aeZJ**, which compares the testing MSE of RLF,GradientBoosting(GB) and XGboost in many real datasets. From the  following table, we see there is a clear advantage of RLF over GradientBoosting(GB) and  RLF is also comparable to XGboost in general.
> > > >
> > > > |Dataset| RLF(100,10,0.8)| GB(1000,0.8)|XGboost(1000,0.8)|
> > > > |-|-|-|-|
> > > > |FF|**4293.74**$\pm 5527.51$|4694.78$\pm 5363.52$|6624.42$\pm 5518.72$|
> > > > |SP|**1.76**$\pm 0.79$|2.13$\pm 0.90$|2.27$\pm 0.90$|
> > > > |EE|0.74$\pm 0.27$| 1.09$\pm  0.22$|**0.09**$\pm 0.03$|
> > > > |CAR|**4916065**$\pm 776998.4$|8781409$\pm 1181857$|5802233$\pm 988893.7$|
> > > > |QSAR|**0.75**$\pm 0.13$|0.85$\pm0.13$|0.97$\pm 0.17$|
> > > > |CCS|23.72$\pm 4.38$|28.87$\pm 3.68$|**20.05**$\pm 8.49$|
> > > > |SOC|422.94$\pm164.72$|460.71$\pm 135.03$|**325.54**$\pm 167.66$|
> > > > |GOM|**240.13**$\pm 28.46$|266.72$\pm 37.29$|269.68$\pm24.96$|
> > > > |SF|0.71$\pm 0.20$|**0.61**$\pm 0.19$|0.95$\pm0.18$|
> > > > |ASN|6.95$\pm 0.79$|20.94$\pm  2.25$|**1.58**$\pm 0.39$|
> > > > |WINER|**0.32**$\pm 0.03$|0.41$\pm 0.038$|0.38$\pm 0.04$|
> > > > |AUV|4324542$\pm 711281.7$|27726483$\pm  1730233$|**155060.7**$\pm 38776.38$|
> > > > |SG|0.013$\pm 0.005$|0.016$\pm  0.005$|**0.012**$\pm0.003$|
> > > > |ABA|**4.60**$\pm 0.58$|4.79$\pm 0.43$|5.76$\pm 0.65$|
> > > > |WINEW|**0.36**$\pm 0.02$|0.50$\pm0.03$|0.40$\pm 0.02$|
> > > > |CPU|**5.45**$\pm 0.49$|6.25$\pm 0.47$|6.00$\pm 1.45$|
> > > > |KRA|0.017$\pm 0.0008$|0.040$\pm 0.0008$|**0.016**$\pm0.0007$|
> > > > |PUMA|**0.00047**$\pm 0.0008$|0.00112$\pm 0.0008$|0.00056$\pm0.0007$|
> > > > |GS|0.00011$\pm 3.81E-06$|0.0003$\pm 9.69E-06$|**8.97E-05**$\pm2.42E-06$|
> > > > |BH|0.01$\pm 0.01$|0.014$\pm 0.008$|**0.004**$\pm0.006$|
> > > > |NPP|**4.81E-07**$\pm 5.12E-08$|3.90E-05$\pm 1.44E-06$| 8.76E-07$\pm8.11E-08$|
> > > > |MH|**0.020**$\pm 0.0009$|0.032$\pm 0.0017$|0.023$\pm0.001$|
> > > >
> > > >
> > > > Lastly, the novelty of our work lies in the invention of a new type of base tree learner, which has the chance of splitting from response Y and exploiting information from response directly.  Most tree ensemble methods (including RF and boosting) only focus on partitioning the X space.   The part of learning directly from responseY  in our work of RLF provides a **new** insight of regression problem, which is consistent with the spirit of the ICLR.
> > > >
> > > > As pointed out by the **reviewer 78aY**, our method shows potential in capturing unobserved confunders, which can be useful when the response has complicated patterns. See the numerical example 2 in section 4.3 of the manuscript.

---

### Comment · Area_Chair_YC8x · 2024-11-13
**authors - reviewers discussion open until November 26 at 11:59pm AoE**

Dear authors & reviewers,

The reviews for the paper should be now visible to both authors and reviewers. The discussion is open until November 26 at 11:59pm AoE.

Your AC

---

> ### Comment · Area_Chair_YC8x · 2024-11-24
>
> Dear reviewers,
>
> The authors have provided individual responses to your reviews. Can you acknowledge you have read them, and comment on them as necessary?
>
> Your AC

---

### Author Response · Authors · 2024-11-13

First of all, we'd like to appreciate the reviewers' interest in our work and we are grateful for their valuable suggestions and feedback. We address each reviewer's specific questions in separate response to their reviews. We hope our additional explanations help to clarify the issues or concerns and hope you find it satisfactory and we are happy to have any further discussions.

In summary, we want to highlight few distinctive characteristics of our RLF.
1.  The introduction of the probability $\tilde{p}$, which controls the type of splitting (Riemann or Lebesgue) at each non-terminal node, makes the RLF more flexible in detecting complex patterns in response Y. Recall that the traditional random forests introduce an extra randomness called "feature bagging", which makes RF different from ordinary bagging ensemble methods. Essentially, the control probability $\tilde{p}$ in RLF provides another randomness in forest methods, which can be interpreted as "splitting type randomness".

2.  In section 4.3, we provided two one-dimensional exmaples illustrating the flexibility of RLF with tuned $\tilde{p}$. More specifically, in datasets with small signal-to-noise ratio (example1), RLF favors more Riemann type splitting. This is intuitive as we typically don't want to split from Y if it's too noisy. On the other hand, when the response Y has mixture distribution (example 2), experiments show that tuned RLF favors more Lebesgue type splitting. This also makes sense as when the response has complicated pattern, the Riemann type cutting might be too simple to capture the pattern of response Y. What's more, when the feature space X has redundant features, i.e sparse model, RLF also prefers more Lebesgue type splitting to avoid learning noise from X space. (See section 4.1)

3.  The performance of RLF in real-world datasets indeed demonstrates the significant superiority over traditional RF. For example, Table 1  shows that RLF with data-driven $\tilde{p}$ outperforms RF in **20** datasets where **eight** of them are significant.  Table 2 shows that RLF with tuned $\tilde{p}$ outperform RF in **23** datasets where **twelve** of them are statistically significant. However, RF only significantly outperforms RLF in two datasets. Table 1 also shows that RLF reduces testing MSE by **15%** on average, which is a decent improvement. Similar improvement can be observed in Table 2. Note that in Table 1 we have **30** datasets in total and Table 2 involves **26** datasets(due to the time efficiency of parameter tuning).

4.  Time complexity of RLF is mainly due to the local random forests and the nature of tree ensemble method. To reduce the time complexity of RLF, we can construct single RLT in parallel and employ a more efficient local model for nodes with Lebesgue splitting, which will be our future work.

We also corrected some typos  in original manuscript where the modified texts are highlighed in blue.  See updated pdf file.

---

### Note · Authors · 2025-01-22

I have read and agree with the venue's withdrawal policy on behalf of myself and my co-authors.